# Efficient and continuous microwave photoconversion in hybrid cavity-semiconductor nanowire double quantum dot diodes

Waqar Khan[1], Patrick P. Potts [2], Sebastian Lehmann [1], Claes Thelander[1], Kimberly A. Dick[1,3], Peter Samuelsson[2] & Ville F. Maisi [1✉]

Converting incoming photons to electrical current is the key operation principle of optical photodetectors and it enables a host of emerging quantum information technologies. The leading approach for continuous and efficient detection in the optical domain builds on semiconductor photodiodes. However, there is a paucity of efficient and continuous photon detectors in the microwave regime, because photon energies are four to five orders of magnitude lower therein and conventional photodiodes do not have that sensitivity. Here we tackle this gap and demonstrate how microwave photons can be efficiently and continuously converted to electrical current in a high-quality, semiconducting nanowire double quantum dot resonantly coupled to a cavity. In particular, in our photodiode device, an absorbed photon gives rise to a single electron tunneling through the double dot, with a conversion efficiency reaching 6%.

[1] NanoLund and Division of Solid State Physics, Lund University, Lund, Sweden. [2] NanoLund and Division of Mathematical Physics, Lund University, Lund, Sweden. [3] Center for Analysis and Synthesis, Lund University, Lund, Sweden. ✉email: ville.maisi@ftf.lth.se

The last two decades have seen an extensive development of photodiodes in the optical regime[1–3], driven by both a need to answer fundamental quantum optics questions and also to develop building blocks of key importance for emergent quantum technologies such as quantum key distribution[4] and linear optics quantum computation[5]. Major developments in the field that show promise to achieve the desired characteristics for such devices and uses include the demonstration of near-unity photon-to-electron conversion efficiency[6] and high operation speed[7,8]. However, these developments have occurred in the optical domain and corresponding developments in the microwave regime are so far largely lacking.

This is a major gap because the realization of an efficient, continuous microwave photodiode would eventually enable extending time-correlated photon counting to the microwave regime and electronics domain. It would also open for quantum technology applications for solid-state system readout, such as photon correlation-based measurements of qubit states[9–11]. But realizing efficient and continuous photon detection in the microwave regime is challenging: while single-shot readout of single-electron tunneling events[12–14] and non-continuous pulsed photon detection via a superconducting qubit[9,10,15–18] have been demonstrated, the continuous and efficient conversion of microwave photons to electric current has so far been missing.

In conventional optical photodiodes, a photon is absorbed by exciting an electron-hole pair over the semiconductor bandgap. The photodiodes' pn-junctions then "split" the electron-hole pairs to generate an electrical current that can be detected. However, in the microwave regime, this approach does not work because the photon energy is four to five orders of magnitude smaller than in the optical regime and suitable semiconductor materials with such small band gaps do not exist. Thus, to realize efficient and continuous photon detection in the microwave regime requires an alternative photodiode operation principle.

Here we present a photodiode device that is capable of efficiently and continuously convert microwave photons into electric current and measure the quantum efficiency of $\eta = 6\%$. Together with the established single-shot detection of electrons[12], our results pave the way for and continuous microwave detection with single-shot readout at the theoretically predicted unit quantum efficiency[19]. The continuous nature of the demonstrated conversion process, similar to the optical photodiodes[1–3], opens up prospects to probe photons and their statistics beyond the gated-time regime[9,10,15–18].

## Results

**Device architecture and operation principle**. Our photodiode device contains a crystal phase-defined double quantum dot (DQD) in a semiconductor InAs nanowire[20] embedded in a superconducting coplanar microwave resonator (Fig. 1). In our approach, the DQD forms an effective electronic two-level system with a tunable energy gap between the ground and excited states acting as an effective bandgap[12,21–23], which is analogous to optical pn-junction-based detectors. During operation, an incident photon enters the cavity from a microwave port and is resonantly absorbed, exciting the DQD[19]. As the ground (excited) state of the DQD is strongly localized on the left (right) dot, photon absorption will result in an electron being transferred from source (left contact) to drain (right contact) with high probability.

**Quantum efficiency**. We realize a maximum efficiency $\eta = 6\%$ for our device. The high efficiency is obtained thanks to the combination of the resonator enhancement of the microwave field in the vicinity of the DQD (which increases the photon-

electron coupling[19]) and the near-unity directivity for the photon-to-electron conversion in the DQD. We obtain this maximum efficiency for a microwave signal incident on the detector, with power $P = 1\,\text{fW}$ and frequency $f = 6.436\,\text{GHz}$, on resonance with both the fundamental resonator mode and the DQD energy gap. This induces a photocurrent $I = 2.4\,\text{pA}$ (see Fig. 2). The photon-to-electron conversion efficiency $\eta$ reads as follows:

$$\eta \equiv \frac{hfI}{eP} = \frac{I/e}{\dot{N}}. \tag{1}$$

Here $\dot{N} = P/hf$ is the rate of incident photons and $I/e$ the rate at which photoelectrons flow from source to drain. The experiment is modeled theoretically within a framework based on the Jaynes–Cummings Hamiltonian (see Methods). We find excellent agreement between experimental data and theory curves, see Figs. 2 and 3.

**Energy detuning dependence**. The electronic and photonic response of the photodetector is shown in the four panels in Fig. 3. In Fig. 3a we present the photocurrent $I$ as a function of the plunger gate voltages $V_{\text{LP}}$ and $V_{\text{RP}}$, applied to the gates LP and RP (see Fig. 1) to move the energy levels of the quantum dots and thus change the level detuning $\delta$ and the energy gap $E = \sqrt{\delta^2 + (2t)^2}$ of the DQD[24]. Here $t$ is the interdot tunnel coupling. For the DQD gap tuned in resonance with the resonator, $E = hf_r$ with $f_r = 6.436\,\text{GHz}$, the photocurrent displays peaks, as expected for photon assisted tunneling in DQDs[21,22]. This occurs at two detunings $\delta = \pm \delta_r$. The two current peaks have the same magnitude but opposite polarity, as anticipated from the symmetry of the DQD-level configurations. Along the lines of constant detuning, $\delta = \pm \delta_r$, the photocurrent remains finite as long as the electrode chemical potential (same for source and drain) is between the ground and excited state of the DQD, giving a peak width $hf_r$. The extension of the photocurrent peaks perpendicular to the constant detuning line is due to resonance broadening and agrees well with the theoretical low power prediction of $\tilde{\Gamma} + 4g^2/\kappa$, where $\tilde{\Gamma}$ denotes the decoherence rate, $g$ the coupling strength of the DQD to the resonator, and $\kappa$ the resonator linewidth.

Similarly to the photocurrent, the photonic response provides information on the DQD-resonator resonance conditions. The phase shift of the resonator transmission as a function of plunger gate voltages is shown in Fig. 3b. The phase shift displays resonances symmetrically at detunings $\delta = \pm \delta_r$, with sharp transitions between positive and negative values, similar to previous DQD-resonator experiments[25–32]. The phase shift resonances arise from interdot transitions without an electron tunneling to source or drain and are hence visible at $\delta = \pm \delta_r$, all along the interdot transition line.

**Frequency response**. The resonator reflection ($R$) and transmission ($T$) coefficients are shown in Fig. 3c as a function of drive frequency $f$ around $f_r$, for detunings $\delta = \delta_r$ and $|\delta| \gg \delta_r$. At $|\delta| \gg \delta_r$, when the DQD is in the Coulomb blockade (CB) regime and no photodetection takes place, the resonance lineshape is very well fitted with a Lorentzian with a central frequency $f_r = 6.436\,\text{GHz}$ and a linewidth of 15.5 MHz [see also Eq. (15) in the Methods section]. For $\delta = \delta_r$, at the positive photocurrent peak, we observe the on-resonance (i.e., when $f = f_r$) transmission decreasing by $\Delta T = -0.06 \pm 0.01$ and the reflection increasing by $\Delta R = 0.02 \pm 0.01$. As is clear from Fig. 3c, these changes are well described by theory. In fact, in the low power limit (see Methods), the active photodetector can be captured by an additional resonator loss term $\kappa_{\text{DQD}} = 4g^2/\tilde{\Gamma}$, decreasing $T$ and increasing $R$ as

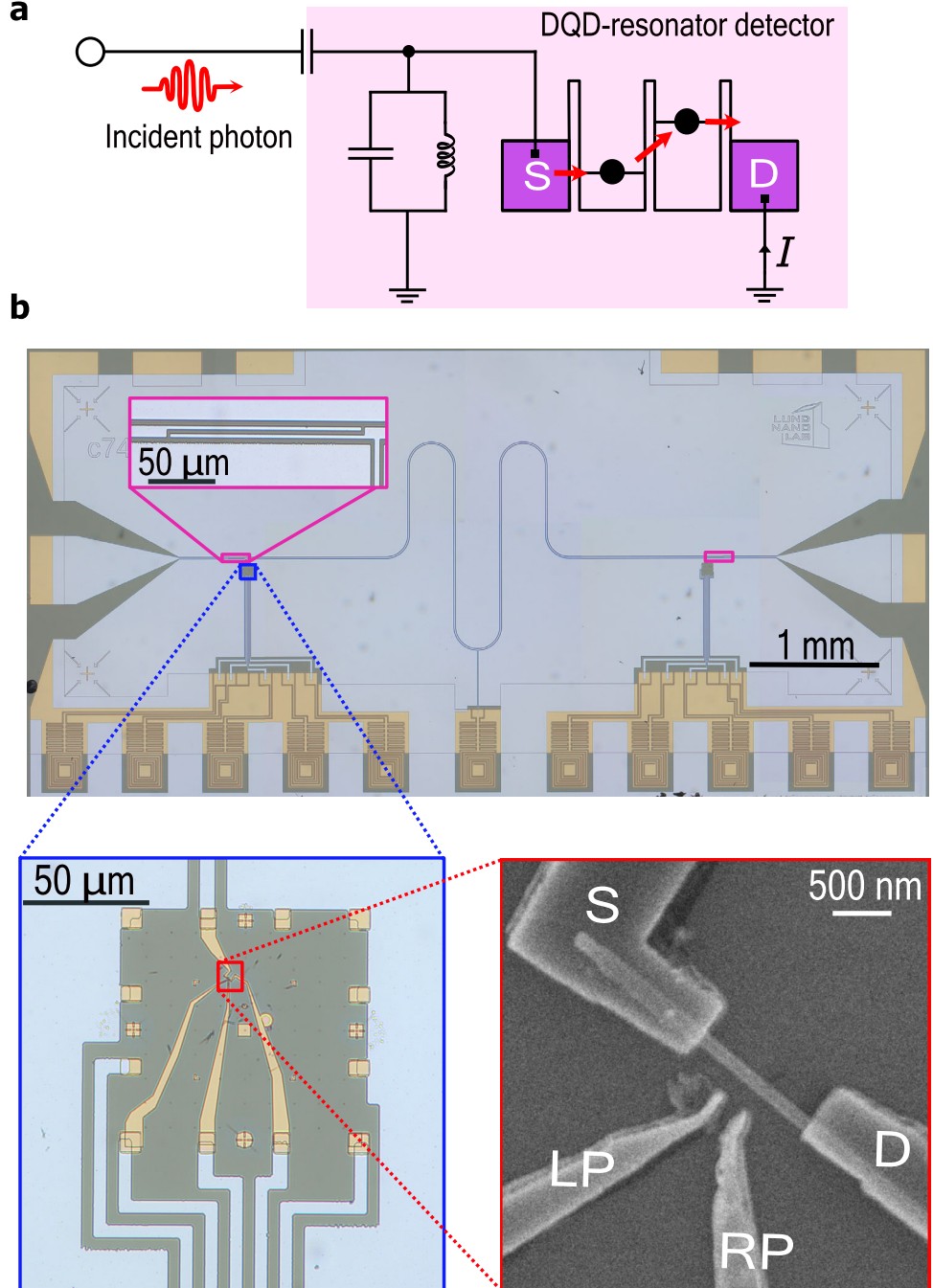

**Fig. 1 Device architecture. a** Schematic of the detector operation: a photon in the transmission line is incident on the detector (pink shaded), consisting of a DQD embedded in a microwave resonator. With large efficiency, the photon is absorbed in the DQD, causing an electron to tunnel through the DQD, producing an electrical current $I$. **b** Optical image of the photodetector highlighting the superconducting microwave resonator made out of aluminum. The two zoom-ins (one optical and one scanning electron micrograph) depict the DQD and its connections to the resonator and measurement lines.

photon absorption by the DQD opens up an additional channel for photons to escape the resonator.

We note that the additional amount of photons removed by the photodetector operation is given by the total change in transmission and reflection, i.e., the DQD decreases the outgoing photon flux by $\Delta T + \Delta R = -4\%$. Interestingly, the amount is comparable to but smaller than $\eta$. The difference arises from a fraction of the photons internally lost in the resonator, which are instead absorbed in the DQD when the detection is switched on, i.e., tuned from CB to the photodetection point: the

photodetection reduces the fraction of photons internally dissipated in the resonator.

Figure 3d shows the photocurrent as a function of drive frequency $f$ for detunings $\delta = \pm\,\delta_r$ and $|\delta| \gg \delta_r$. At $\delta = \pm\,\delta_r$ the photocurrent lineshape, for both positive and negative current peaks, is within measurement accuracy the same as the one of the resonator response in Fig. 3c. We note that the detector bandwidth is given by the resonance linewidth. Indeed, in the low power limit, theory predicts a Lorentzian with the same central frequency and width as for the transmission. This

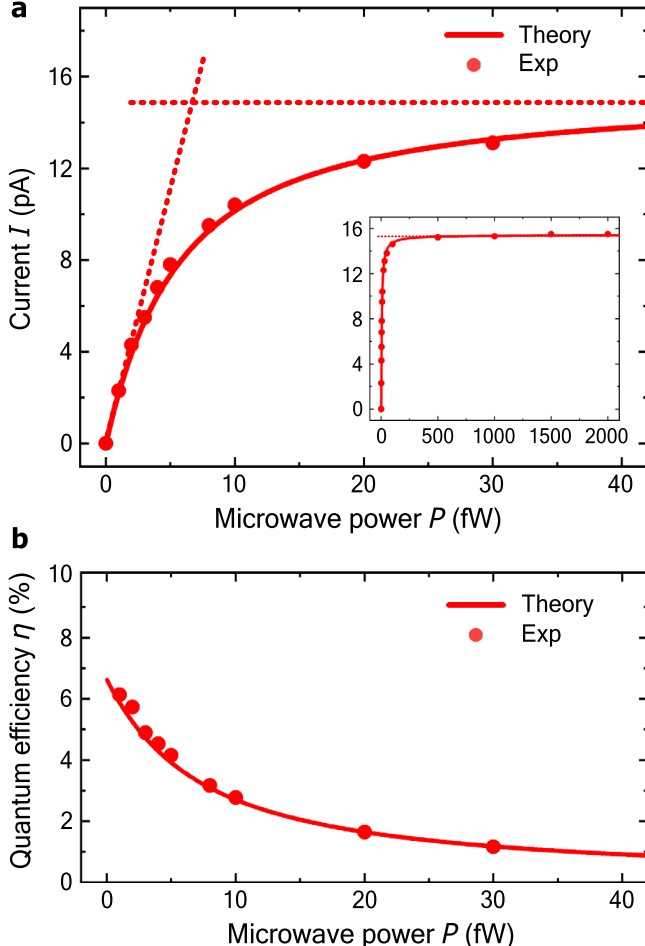

**Fig. 2 Photoresponse and quantum efficiency. a** Photocurrent $I$ as a function of the ingoing microwave power $P$. Dots (solid line) present the measured data (theory) and the dashed lines are the theoretical predictions for the low power, linear response current (with $\eta = 6.6\%$) and the high power saturation current $I = 15$ pA. The incident microwave drive signal is in resonance with the resonator, $f = f_r$, and the DQD, $E = hf$, at a detuning $\delta = + \delta_r$ (c.f. Fig. 3). No bias voltage is applied. **b** Quantum efficiency, Eq. (1), determined from the data of panel **a**. At 1 fW input power, we experimentally reach 6% efficiency. Exp: experimental data.

concurrence between the photocurrent and resonator response lineshapes provides direct evidence that the photodetector signal arises when photons entering the resonator are absorbed by the DQD, giving rise to an electrical current. At $|\delta| \gg \delta_r$, for the DQD in the CB regime, no photocurrent is observed as expected since the photodetector is tuned away from the operation point.

**Theoretical modeling**. The detector parameters are determined by fitting the measured results to the predictions of the full theory. As a first step, the bare resonator properties are determined with the DQD in the CB regime. By fitting the full frequency dependence of the reflection and transmission coefficients, in Fig. 3c, the left and right resonator port couplings and the internal resonator losses are determined to $\kappa_L/2\pi = 4.3$ MHz, $\kappa_R/2\pi = 5.4$ MHz, and $\kappa_{int}/2\pi = 5.8$ MHz, respectively. This gives the linewidth $\kappa = \kappa_R + \kappa_L + \kappa_{int} = 2\pi \cdot 15.5$ MHz.

Next, for the DQD properties, we start by considering the directivity $p_f$, which is the fraction of photoelectrons that traverse the DQD from left to right, minus the fraction that traverse the DQD from right to left. For symmetric tunnel rates to source and drain, we find $p_f = \delta_r/hf_r$. We determine $\delta_r$ and the interdot

tunnel coupling $t$ from the resonance conditions of Fig. 3b. The fit yields $t = 8.1$ μeV and $\delta_r = 21$ μeV, resulting in $p_f = 0.8$. Then, the tunnel rate $\Gamma/2\pi = 46$ MHz is obtained from the high power saturated photocurrent $I = 2e\Gamma p_f/5$, see Fig. 2. This tunnel rate also characterizes the dead time of the detector $\sim \Gamma^{-1} = 20$ ns.

Finally, by fitting the data shown in Fig. 3b–d, we extract a DQD-resonator coupling of $g/2\pi = 21$ MHz (corresponding to a bare coupling of $g_0/2\pi = 38$ MHz, see Methods) influencing predominantly the strength of the phase response of Fig. 3b, the total decoherence rate of $\tilde{\Gamma}/2\pi = 790$ MHz, influencing its smearing, and the DQD relaxation rate of $\Gamma_r/2\pi = 23$ MHz that, in addition to the other parameters, determines the low power photocurrent and the quantum efficiency of Fig. 2. With these parameters, we find excellent quantitative agreement between theory and measurements including both the resonator response as well as the photocurrent.

## Discussion

To analyze the possibilities for further improving the photodetector performance, we note from Fig. 2 that the quantum efficiency is maximal in the low microwave power regime. Theory gives, in line with ref. [19], the low power efficiency:

$$\eta = \frac{\kappa_L}{\kappa} \frac{4\,\kappa_{DQD}\,\kappa}{(\kappa_{DQD} + \kappa)^2} \frac{\Gamma}{\Gamma + \Gamma_r}\, p_f. \qquad (2)$$

Hence, $\eta$ approaches unity when four conditions are met. First, the linewidth $\kappa$ needs to be dominated by the input port: $\kappa = \kappa_L$. Second, the effective rate at which photons are absorbed by the DQD, $\kappa_{DQD}$, should match the resonator linewidth: $\kappa = \kappa_{DQD}$. Since $\kappa_{DQD} = 4g^2/\tilde{\Gamma}$, a larger coupling constant $g$ would allow unit efficiency with higher bandwidth (larger $\kappa$) and/or higher decoherence rate $\tilde{\Gamma}$. Third, the tunneling rate $\Gamma$ should be much larger than the DQD relaxation $\Gamma_r$. Fourth, the tunneling between the dots $t$ should be small compared to the photon energy $hf_r$ to obtain $p_f \rightarrow 1$. The second and the fourth condition have a trade-off because a small interdot tunnel coupling suppresses interaction with the resonator as $g \propto t$ [30], and therefore reduces $\kappa_{DQD}$. Unit efficiency is thus approached as the detector slows down. We estimate, from Eq. (2), that with an order of magnitude higher quality factor and source-drain tunnel coupling $\Gamma$, and a one-port cavity, $\kappa_L = \kappa = \kappa_{DQD} = 2\pi \cdot 2$ MHz, our photodetector would immediately reach a quantum efficiency of $\eta = p_f = 80\%$. Based on previous work on microwave resonators interacting with quantum dots, such high-quality factor resonators and larger source-drain tunnel rates $\Gamma/2\pi \sim 1$ GHz are experimentally obtainable [25,27,29] by adjusting the geometry of the input coupling capacitance and minimizing the internal losses of the resonator. This thus makes near-unity quantum efficiency detectors experimentally accessible.

Comparing our experiment to previous works, we demonstrated here continuous photodetection with high efficiency in the microwave domain thanks to the combination of the resonator enhancement of the microwave field in the vicinity of the DQD, increasing the photon-electron coupling [19], and the high-quality polytype DQDs with atomically sharp interfaces enabling a near-unity directivity for the photon-to-electron conversion. In addition, and of key importance for photocounting applications, the photodiode operation is continuous in time; a detected photon directly gives rise to an electron transfer through the DQD. This is in stark contrast to state-of-the-art single microwave photon detectors based on superconducting qubits [9,10,15–18], where photodetection, despite of reaching close to unity efficiency and demonstrating non-demolition capability, is indirect, occurring via qubit readout only at predetermined times.

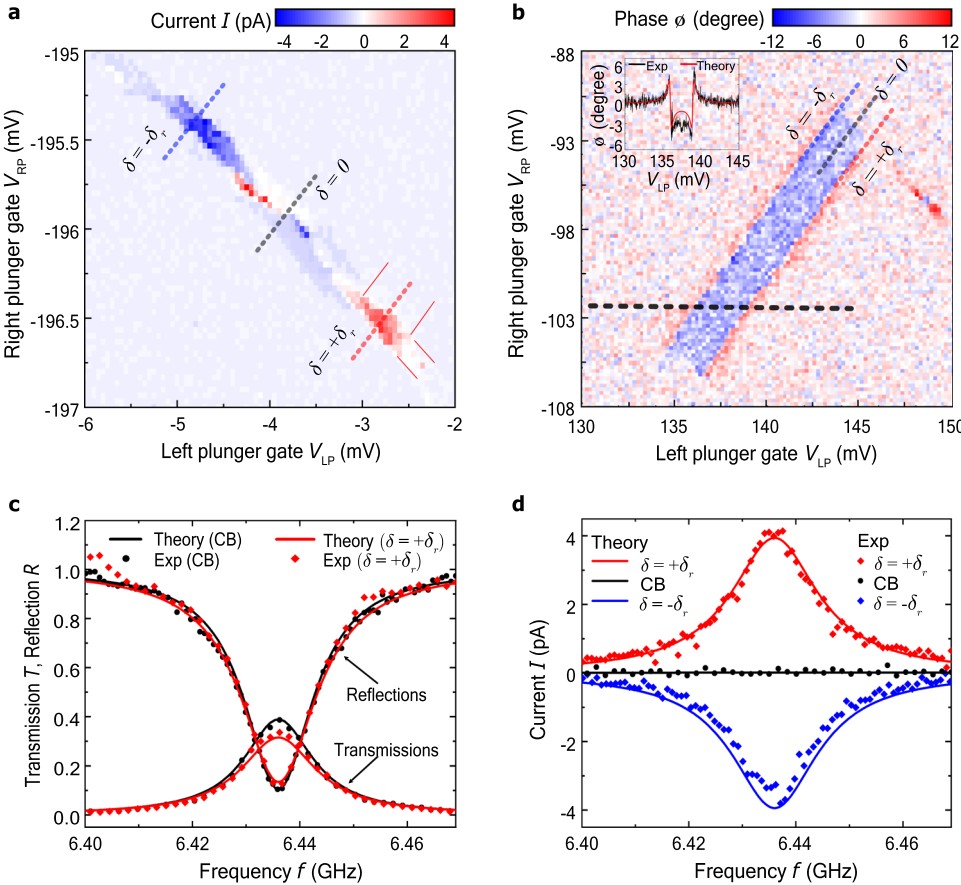

**Fig. 3 Electronic and photonic response. a** Photocurrent $I$ as a function of the plunger gate voltages $V_{LP}$ and $V_{RP}$. The photocurrent shows peaks with opposite signs at detunings $\delta = \pm \delta_r$. The width of the peaks parallel and perpendicular to the lines of constant detuning (shown as dashed lines at $\delta = \pm \delta_r$, 0) is given by the DQD energy gap $hf_r$ and $\tilde{\Gamma} + 4g^2/\kappa = 0.9$ GHz respectively, as predicted by theory (thin solid lines at positive current peak). **b** Phase shift $\phi$ of the resonator transmission as a function of $V_{LP}$ and $V_{RP}$, with constant detunings $\delta = \pm \delta_r$, 0 shown as thin dashed lines. Inset shows cross-section at $V_{RP} = -102$ mV (along thick dashed line in main panel) together with theoretical fit. **c** Transmission and reflection coefficients as a function of drive frequency $f$ at $\delta = \delta_r$ (red symbols, active photodetection) and for the DQD in the Coulomb blockade (CB) regime, at $|\delta| \gg \delta_r$ (black symbols, no photodetection). **d** Measured photocurrent as a function of $f$, together with theoretical fit, at $\delta = \pm \delta_r$ and at $|\delta| \gg \delta_r$ (CB regime). In all panels, the microwave power was $P = 2$ fW and no source-drain voltage was applied. The measurements in panels **a** and **b** were made at resonance $f = f_r$. The plunger gate voltages in panels **c** and **d** are $V_{LP} = -2.8$ mV and $V_{RP} = -196.6$ mV for $\delta = \delta_r$ and $V_{LP} = -4.7$ mV and $V_{RP} = -195.4$ mV for $\delta = -\delta_r$. Exp: experimental data.

Previous continuous detectors have predominantly focused on microwave spectroscopy and have an unknown, presumably orders of magnitude lower efficiency[12,21–23]. In particular, the detection of single absorption events reported in ref. [12] obtains a photoelectron only for ~$10^{-7}$% of the generation events of the QPC emitter. This is seven orders of magnitude smaller than our 6% efficiency. Furthermore, we unambiguously detect the photons that impinge on the detector while ref. [12] potentially features alternative channels such as phonons and plasmons.

Moreover, the efficiency demonstrated here is two orders of magnitude higher than the one for photoemission in similar hybrid cavity structures[26,33]. The low emission efficiency is explained by the dominance of DQD relaxation over photoemission, occurring at rates $\Gamma_r$ and $\kappa_{DQD}$, respectively. For photodetection, the DQD relaxation instead competes with tunneling out of the DQD (rate $\Gamma$) that occurs at comparable rates in our device.

In conclusion, we have experimentally demonstrated conversion of microwave photons to electrical current and with that constructed a microwave photodetector. Combining our results with a standard single-shot charge readout together with high-quality resonators opens up the avenue to build and perform microwave experiments with individual photons with high quantum efficiency.

## Methods

**Device fabrication.** The microwave resonator is realized with superconducting Al/Ti layer on undoped Si/SiO$_2$ substrate with 200-nm thick SiO$_2$, using conventional photolithography techniques. The two ends of the resonator connect capacitively to microwave ports from which we send in photons as well as measure the outgoing photons. The DQDs were formed in InAs zinc-blende–wurtzite heterostructures grown with metal organic vapor phase epitaxy. The growth process is described in ref. [34] and the integration of the DQD to the device was done by following the recipe of ref. [20]. Gold pads were used as interconnects to obtain conducting contacts between the resonator and the DQD lines. The coupling of the resonator and the DQD was achieved by connecting the voltage antinode of the resonator capacitively to the dipole moment of the interdot transition via the source electrode of the DQD[27]. This connection allows for grounding the DQD at the middle point of the resonator. The middle point of the resonator has a voltage node point and hence the connection does not distort the resonance[27,29] while allowing for a finite photocurrent to flow through the structure without generating bias voltage and applying a bias voltage for the transport characterization.

The device fabrication was made by starting with two photolithography steps to fabricate metallic DC and interconnect pads of Ni/Au with thickness 5 nm / 95 nm and microwave resonator from Ti/Al with thickness 5 nm / 200 nm. After fabricating the resonator, the process continued as in ref. [20]: InAs nanowires were transferred onto the designated spots with pre-patterned alignment marks by pick and drop method. In the next step, electron-beam lithography was opted to connect one side of nanowire with the microwave resonator that serves as source reservoir and the other end of nanowire is connected to a metallic DC pad, which acts as drain reservoir. In addition to source/drain contacts, gate electrodes are also

formed using electron-beam lithography techniques. The electrode material for contacts is Ni with thickness 135 nm.

**Measurements**. The measurements were performed in a dilution refrigerator at a base temperature below 10 mK and electronic temperature of 40 mK. The measurements were started by characterizing the DQD with standard transport measurements (Supplementary Fig. 1a) and the resonator by measuring transmission from left to right microwave port (Fig. 3d, CB). The resonator modes, i.e., frequency $f_r$ and linewidth $\kappa$, were determined from the transmission data. The tunnel couplings of the DQD were then tuned by choosing the electron numbers with the plunger gate voltages $V_{LP}$ and $V_{RP}$ such that the current in the cotunneling lines (marked gray in Supplementary Fig. 1a) was of equal magnitude for the source and drain transition and the interdot coupling is small based on the phase response of Fig. 3b. These two conditions are obtained by tuning the difference and the mean of $V_{LP}$ and $V_{RP}$, respectively. After that, the measurements of the main article were performed with simple sweeps. Shifts in the voltages $V_{LP}$ and $V_{RP}$ arising from offset charge changes were compensated in the course of the measurement to keep the photodetector in the same operation point.

To calibrate ingoing and outgoing microwave power, we used the reflected power away from the resonance as a reference. In this case all the power is reflected away from the resonator allowing us to determine the cable losses and thus determine the ingoing and outgoing powers and transmission and reflection coefficients. The Supplementary Methods present the used microwave lines and lists the insertion losses and gains of the microwave components and describes the protocol and calculations of the calibration.

We determine the transmission coefficient presented in the main article from the left port to right one, and the reflection coefficient by sending in a signal via microwave circulator to the right port and measuring it with the same amplification chain. Based on the reciprocity of the resonator response, the transmission from left to right port is identical to the reverse direction. Hence, these measurements present the reflection and transmission for a signal sent to the right port. Note that this conclusion holds even for a very asymmetric resonator. In addition, we have a rather symmetric system with $\kappa_L \approx \kappa_R$.

To determine the directivity $p_f$ of the photodetector, we determined the detuning of the two resonance conditions $\delta = \pm \delta_r$ of Fig. 3b. We used the finite bias measurement of Supplementary Fig. 1a as a reference to determine the lever arm for the detuning $\delta$ in response to the gate voltages. The calculations are presented in the Supplementary Methods.

The linewidths of the data in Fig. 3c, d are:

- Transmission in CB: $15 \pm 1$ MHz
- Reflection in CB: $17 \pm 2$ MHz
- Transmission, photodetection: $16 \pm 1$ MHz
- Reflection, photodetection: $17 \pm 3$ MHz
- Photocurrent, pos. polarity: $17 \pm 3$ MHz
- Photocurrent, neg. polarity: $19 \pm 3$ MHz

The linewidths and their uncertainty were determined by fitting a Lorentzian lineshape to the data. The fits yield the same linewidth within the fit uncertainty. All of the fits provide also the same resonance frequency of 6.436 GHz within 1 MHz uncertainty.

**Theory**. Theoretically, we describe the system using a Jaynes–Cummings Hamiltonian, given here in the frame rotating at the angular frequency of the incoming radiation $\omega$ (here $\hbar = 1$):

$$\hat{H} = \Delta_c \hat{a}^\dagger \hat{a} + \Delta_q \frac{\hat{\sigma}_z}{2} + g(\hat{a}\hat{\sigma}^\dagger + \hat{a}^\dagger \hat{\sigma}) + \sqrt{\kappa_{in}\dot{N}}(\hat{a}^\dagger + \hat{a}), \quad (3)$$

where $\Delta_c = \omega_r - \omega$, with the angular frequency $\omega_r = 2\pi f_r$, and $\Delta_q = \sqrt{\delta^2 + 4t^2} - \omega$, where $\delta$ denotes the difference in the on-site energies of the left and right dot. The Jaynes–Cummings coupling can be written in terms of the bare coupling as $g = g_0 2t/\omega_r$. The resonator is described by creation (annihilation) operators $\hat{a}^\dagger$ ($\hat{a}$), $\dot{N}$ denotes the rate of incident photons, and $\kappa_{in}$ the loss rate of the input port. The DQD is described by a three-level system provided by the bonding (ground) state $|g\rangle$, the anti-bonding (excited) state $|e\rangle$, each containing one extra electron in the DQD, and the empty state $|0\rangle$ where no extra electron resides in the DQD. Double occupancy of the DQD is prevented by Coulomb interactions. The spin matrices act on the subspace of ground and excited state, i.e., $\hat{\sigma}_z = |e\rangle\langle e| - |g\rangle\langle g|$ and $\hat{\sigma} = |g\rangle\langle e|$. The empty state has an energy equal to the chemical potential of the contacts, which is set to zero. We note that Eq. (3) is only valid for detunings close to the resonance value $\delta_r$ due to a rotating wave approximation.

Tunneling of electrons between the reservoirs and the DQD, photon losses, relaxation, and decoherence are described by the Markovian master equation:

$$\partial_t \hat{\rho} = -i[\hat{H}, \hat{\rho}] + \Gamma \mathcal{D}[|0\rangle\langle e|]\hat{\rho} + 2\Gamma \mathcal{D}[|g\rangle\langle 0|]\hat{\rho} + \Gamma_r \mathcal{D}[\hat{\sigma}]\hat{\rho} + \Gamma_\phi \mathcal{D}[\hat{\sigma}_z]\hat{\rho} + \kappa \mathcal{D}[\hat{a}]\hat{\rho}, \quad (4)$$

with the superoperator $\mathcal{D}[\hat{A}]\hat{\rho} = \hat{A}\hat{\rho}\hat{A}^\dagger - \frac{1}{2}\{\hat{A}^\dagger \hat{A}, \hat{\rho}\}$. Here $\Gamma$ denotes the tunnel rate between source/drain and DQD (assumed equal for source and drain), $\Gamma_r$ is an internal relaxation rate (presumably induced by phonons), $\Gamma_\phi$ a decoherence rate

(presumably induced by coupling to nearby charges), and $\kappa$ denotes the photon loss rate. Spin degeneracy is taken into account by the factor of two in the term that describes electrons entering the DQD.

An electron that leaves the DQD upon absorbing a photon, i.e., a photoelectron, may both originate and end up either in the left or right contact. The corresponding tunnel rates are given by:

$$\Gamma_R^{out} = \frac{\Gamma_L^{in}}{2} = \Gamma \frac{\omega_r + \delta_r}{2\omega_r}, \quad \Gamma_L^{out} = \frac{\Gamma_R^{in}}{2} = \Gamma \frac{\omega_r - \delta_r}{2\omega_r}, \quad (5)$$

where the terms $(\omega_r \pm \delta_r)/2\omega_r$ account for the localization of the wave functions, and the factor of two difference between in and out tunneling accounts for spin degeneracy. Note that since $\omega_r \gg k_B T$, electrons may only enter the DQD into the ground state and leave the DQD from the excited state. The charge current through the DQD may then be written as follows:

$$I = \langle e|\hat{\rho}|e\rangle e \Gamma_R^{out} - \langle 0|\hat{\rho}|0\rangle e \Gamma_R^{in}, \quad (6)$$

It is instructive to introduce the directivity, defined as the fraction of photoelectrons entering from the left and leaving to the right, minus the fraction entering from the right and leaving to the left:

$$p_f = \frac{\Gamma_L^{in}\Gamma_R^{out}}{2\Gamma^2} - \frac{\Gamma_R^{in}\Gamma_L^{out}}{2\Gamma^2}. \quad (7)$$

For symmetric tunnel rates, as assumed here, the directivity reduces to the simple expression $p_f = \delta_r/\omega_r$.

In the low-drive limit $\dot{N} \to 0$, the density matrix may be expanded perturbatively to lowest order in $\sqrt{\dot{N}\kappa_{in}}$. This results in the current:

$$I = \frac{e\dot{N}16g^2\kappa_L\tilde{\Gamma}p_f\Gamma/(\Gamma + \Gamma_r)}{16g^4 + 8g^2(\kappa\tilde{\Gamma} - 4\Delta_c\Delta_q) + (\tilde{\Gamma}^2 + 4\Delta_q^2)(\kappa^2 + 4\Delta_c^2)}, \quad (8)$$

where we chose the left port as the input, $\kappa_{in} = \kappa_L$. From this equation, the expression for the efficiency given in Eq. (2) in the main text is recovered by setting $\Delta_c = \Delta_q = 0$. For $\Delta_c = 0$, Eq. (8) reduces to a Lorentzian (as a function of $\Delta_q$) with width $\tilde{\Gamma} + 4g^2/\kappa$. This explains the width of the photocurrent peaks in Fig. 3a. For $\delta = \delta_r$ (i.e., $\Delta_c = \Delta_q = \omega_r - \omega$), and in the limit $\tilde{\Gamma} \gg \kappa + \kappa_{DQD}$ [with $\kappa_{DQD} = 4g^2/\tilde{\Gamma}$], Eq. (8) reduces to a Lorentzian with width $\kappa + \kappa_{DQD}$.

In the large drive regime, we can assume that the backaction of the DQD on the cavity, as well as fluctuations of the cavity field can be neglected and we replace:

$$\hat{a} \to -\frac{2i\sqrt{\dot{N}\kappa_L}}{\kappa}. \quad (9)$$

This results in the current:

$$I = e\frac{16g^2\kappa_L\dot{N}\Gamma p_f}{\kappa^2\tilde{\Gamma}(\Gamma + \Gamma_r) + 40g^2\kappa_L\dot{N}}. \quad (10)$$

Equation (10) saturates at large drives to the value:

$$I|_{\dot{N}\to\infty} = e\frac{2\Gamma}{5}p_f. \quad (11)$$

The theory curves shown in Figs. 2 and 3d (at $\delta = \delta_r$) are obtained using Eq. (6) and solving the master equation numerically with $\kappa_{in} = \kappa_L$, except for the inset of Fig. 2a where Eq. (10) was used to avoid dealing with large Hilbert space dimensions. The photocurrent at $\delta = -\delta_r$, is given by the negative of the current at $\delta = \delta_r$ due to symmetry.

To model the transmission and reflection coefficients, we use the input-output relations:

$$\langle \hat{b}_{in,\alpha}\rangle + \langle \hat{b}_{out,\alpha}\rangle = \sqrt{\kappa_\alpha}\langle \hat{a}\rangle, \quad (12)$$

with $\alpha = L, R$. When the right port is used as the input, we obtain the reflection and transmission amplitudes:

$$\begin{aligned} r &= \frac{\langle \hat{b}_{out,R}\rangle}{\langle \hat{b}_{in,R}\rangle} = i\sqrt{\frac{\kappa_R}{\dot{N}}}\langle \hat{a}\rangle - 1, \\ t &= \frac{\langle \hat{b}_{out,L}\rangle}{\langle \hat{b}_{in,R}\rangle} = i\sqrt{\frac{\kappa_L}{\dot{N}}}\langle \hat{a}\rangle, \end{aligned} \quad (13)$$

where we used $\langle \hat{b}_{in,R}\rangle = -i\sqrt{\dot{N}}$ and $\langle \hat{b}_{in,L}\rangle = 0$. The theory curves shown in Fig. 3b, c are obtained from Eq. (13) and solving the master equation numerically with $\kappa_{in} = \kappa_R$.

In the low-drive limit at $\delta = \delta_r$ and under the assumption $\tilde{\Gamma} \gg \kappa + \kappa_{DQD}$ we find:

$$\begin{aligned} T \equiv |t|^2 &= \frac{\kappa_L\kappa_R}{\left(\frac{\kappa}{2} + \frac{\kappa_{DQD}}{2}\right)^2 + (\omega - \omega_r)^2}, \\ R \equiv |r|^2 &= \frac{\left(\frac{\kappa}{2} + \frac{\kappa_{DQD}}{2} - \kappa_R\right)^2 + (\omega - \omega_r)^2}{\left(\frac{\kappa}{2} + \frac{\kappa_{DQD}}{2}\right)^2 + (\omega - \omega_r)^2}. \end{aligned} \quad (14)$$

We note that the transmission $|t|^2$ shows the same Lorentzian lineshape as the photocurrent in this limit.

In the CB regime (i.e., $|\delta| \gg \delta_r$), the DQD and the cavity are effectively decoupled and we model this regime by setting $g = 0$ in Eq. (3). We find the standard expressions for the transmission and reflection coefficients:

$$
\begin{aligned}
T &= \frac{\kappa_L \kappa_R}{\left(\frac{\kappa}{2}\right)^2 + (\omega - \omega_r)^2}, \\
R &= \frac{\left(\frac{\kappa}{2} - \kappa_R\right)^2 + (\omega - \omega_r)^2}{\left(\frac{\kappa}{2}\right)^2 + (\omega - \omega_r)^2}.
\end{aligned}
\tag{15}
$$

## Data availability
The data that support the findings of this study are available from the authors upon reasonable request.

## Code availability
The numerical calculations were performed using QuTiP[35].

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

## Acknowledgements
We thank P. Krantz for useful discussions and acknowledge the financial support from NanoLund, Swedish National Science Foundation, and the Knut and Alice Wallenberg Foundation through the Wallenberg Center for Quantum Technology (WACQT). P.P.P. acknowledges funding from the European Union's Horizon 2020 research and innovation programme under the Marie Skłodowska-Curie Grant Agreement No. 796700.

## Author contributions
V.F.M. conceived the experiment. S.L., C.T. and K.D.T. designed and fabricated the nanowires. V.F.M. and W.K. designed and fabricated the nanowire-cavity device. W.K. performed the measurements. W.K., V.F.M., P.P.P. and P.S. performed the data analysis. P.P.P. performed the theoretical calculations and the numerical simulations. All authors contributed to the discussion of the results and the manuscript preparation.

## Funding

## Competing interests
The authors declare no competing interests.
