## [Peer Review File · Nature Communications]

Reviewers' Comments:

Reviewer #1:

Remarks to the Author:

In their manuscript, the authors claim the efficient and continuous detection of microwave photons demonstrating the functionality of a photon detector implemented in a nanowire double quantum dot (DQD) diode coupled to a superconducting coplanar resonator. They fabricated a crystal phase defined nanowire DQDs in correspondence of the electric antinode of the resonator and measured the source-drain current which, in resonant conditions, is attributed to a photon-assisted tunneling process between DQD levels. By measuring magnitude and phase of reflection and transmission coefficients of the resonator they also obtain additional information on level spacing and photon absorption in the DQD detector.

Microwave photon detectors based on DQDs were reported a long time ago but, as the authors correctly pointed out in their manuscript, the efficiency of such detectors was very low. In recent times, there has been a growing interest in the development of suitable methods for the detection of single microwave photons. While important results have been obtained with superconducting circuits, semiconductor DQD-based detectors have lagged behind. The working principle of the latter is basically photon-assisted tunneling in DQD devices, which is not new for its own sake, nor is the hybrid DQD-resonator architecture. However, experiments that define conditions for developing efficient microwave photon detectors deserve attention. This manuscript certainly covers a very interesting topic and I think it could be of interest for a broad audience that extends beyond the already large communities of semiconductor devices and hybrid quantum systems. There are however a series of points in which the reported results are not convincing, thus I have serious concerns regarding the soundness of their claims. Several experimental details are missing and the readability of the manuscript should also be improved. In the following I list my comments and questions, which must be addressed before I can recommend the manuscript for publication.

1. In the abstract, the authors claim the efficient and continuous detection of "itinerant" photons by the nanowire double quantum dot. Strictly speaking this is not correct: in their experiment, the detector probes the photons stored in a cavity, not directly the itinerant photons entering from the transmission line. This point has been stressed in different experiments with superconducting circuits [see e.g. Nature Physics 6, 663 (2010), Nature Physics 14, 546 (2018) and Phys. Rev. X 8, 021003 (2018)]. The authors should correct their claim.
2. In Fig. 2 photocurrent data are presented as a function of the microwave power. How many photons are on average in the resonator? Could the authors evaluate the conversion factor between the measured current and the photon number?
3. For the data in Figure 2, bias and gate voltages are not indicated, thus the experimental conditions are not clear. In addition, details on the theoretical curves are not given, the reader has to scroll down to the end of the paper in the Methods section in order to find information on the fit. A more detailed description of Figure 2 would be appreciated.
4. In my opinion, it is odd to show the Fig. 2 (photoresponse and quantum efficiency as a function of power) before Fig. 3 (electronic and photonic response as a function of gate voltages and frequency). I think the other way round would be clearer.
5. Page 2, left column. The authors state that a photocurrent is observed when the energy gap of the double quantum dot (DQD) matches the photon energy. The numerical values of inter-dot tunneling coupling (t) and detuning (Δ_r) are given only in the supplemental material. I recommend indicating these numbers also in the manuscript in order to strengthen the analysis of the experimental results.
6. In Methods section the authors state that "Shifts in the voltages VLP and VRP arising from offset charge changes were compensated in the course of the measurement to keep the photodetector in the same operation point." This behavior suggests that the charge configuration of the nanowire DQD is not stable. I wonder how the authors can safely assert that the operation point is the same after the reconfiguration of the DQD. Could the authors clarify this point?
7. The stability diagram is shown only as supplemental material and for a range of plunger gate voltages different from those in Fig. 3a and b. The configuration of the double quantum dot for the data in Fig. 3 is not clear. What is the source-drain bias voltage? Is there an estimation of the electron occupancy? Fig. 3a and 3b have different values of the plunger gate voltages. Direct comparison of data taken in different conditions is misleading, the authors should explain why measurements are not taken in the same conditions. What is the voltage of the plunger gates in

Fig. 3c and d?

8. In Fig. 3a, the photocurrent peaks are relatively large spots along the detuning axis. What are the factors that determine such a large broadening? Why there is a measurable photocurrent when resonator and DQD are not resonant ($\delta \neq \pm \delta_r$)? Additionally, in Fig. 3a the reversal of the current polarity is visible for δ near zero. Why?

9. I am puzzled by Fig. 3a. The photon-assisted tunneling process typically gives rise to current peaks when the spacing between left and right dot levels matches the microwave frequency. Once that the detuning is fixed, the resonance condition can be satisfied also by slightly shifting the levels of both dots with respect to the leads. This is what is observed in the phase measurement in Fig. 3b. Accordingly, I would expect that the photocurrent forms two "ridges" aligned along the normal of the detuning axis: this is what is typically observed in microwave spectroscopy of DQDs [see e.g. Rev. Mod. Phys. 75, (2003), pages 14-17]. However, Fig. 3a is in stark contrast with this behavior: nonzero current is observed along the detuning axis, while in the normal direction the current is null. Why? Does it mean that the spacing between DQD levels and the leads matters, thus there is an effect of the leads? I think this is a crucial point, the authors should explain this discrepancy between photocurrent, phase data and the literature. A schematic diagram of the DQD levels would be useful in order to explain the tunneling process related to Fig. 3a.

10. Data in Fig. 3a are taken with microwave power of 2 fW, control data taken without external microwave drive are not shown. Is there a background current? Could it be used to quantify the dark count rate of the detector?

11. Could the authors exclude the presence of excited states within the bias window? There seems to be a non-constant current in the bias triangles shown in Fig. 1a of supplemental material.

12. On page 2, while commenting on Fig. 3b the authors state that "The dispersive phase shift of the resonator transmission displays also the two resonances $\delta = \pm \delta_r$ ". This sentence is a little clumsy and needs to be reformulated. The dispersive limit takes place for large detunings, while for $\delta = \pm \delta_r$ the resonator and the DQD are on resonance.

13. On page 2, while commenting on Fig. 3c and d, the authors state that "the photocurrent follows the same Lorentzian lineshape as the resonator response". The fitted values of resonance frequency and linewidth are not reported. Could the authors be more specific on this point? What is the frequency selectivity, hence the bandwidth, of the detector?

14. On Page 3 it is stated that "We observe the transmission decreasing by $\Delta T = -0.06$ and the reflection increasing by $\Delta R = 0.02$ ". This variation is relatively small, how wide are the error bar?

15. On Page 3 below in the same paragraph it is stated that "This reduction exemplifies that the DQD absorption removes photons from the cavity and it is comparable to but interestingly smaller than η ." It would be useful to check this behavior also at different microwave power, thus different efficiency.

16. On page 3, left column, the authors state that "The effect of the DQD is captured by a change in linewidth given by an additional loss term $\kappa_{DQD} = 4g^2/\Gamma$ ". However, g and Γ are not defined at this point. In the same paragraph, the detection is said to be "switched on". Do the authors mean that the double quantum dot is polarized with detuning equal to δ_r ?

17. The total decay rate of the resonator is $\kappa = \kappa_L + \kappa_R + \kappa_{int} = 15.5$ MHz, it thus follows that the loaded quality factor is $Q = f_r/\kappa = 415$. This value is significantly lower than what reported in the literature. Large κ_L and κ_R follows as a consequence of the large capacitances between resonator and feed lines. However, the internal loss rate is quite high, even with LC filters included in the dc lines to suppress microwave leakage. Do the authors have an explanation for this behavior?

18. Page 4, left column. "Based on previous work on microwave resonators interacting with quantum dots [17, 19, 21]..." Several references are missing, see Sensors 20, 4010 (2020).

19. Page 4, left column. The coupling strength is said to be proportional to the tunneling rate. A reference would be useful here.

20. In ref. Phys. Rev. A 95, 012325 (2017), the efficiency at the optimal point is obtained by tuning inter-dot tunnel coupling, DQD-resonator coupling strength and total decay rate of the resonator. The tunneling rate between dots and leads also plays a role. The efficiency determined by Eq. 2 does not seem to be directly dependent on such parameters. Could the authors better explain these differences with the model by Wong and Vavilov?

Reviewer #2:

Remarks to the Author:

In this work, the authors demonstrated high-efficiency conversion of microwave photons into an electrical current, using a double quantum dot as an absorber of single microwave photon. From the slope of the current-microwave power plot, they concluded the efficiency of conversion reaches 7%. The electronic and photonic responses (Figure 3) support this estimation. From these data, the authors argue that this is the first realization of continuous photodetection in the microwave domain.

However, in my opinion, the authors' claim that this is the first continuous microwave photodetection is too strong. The authors should also demonstrate that the electric current induced by a single microwave photon can be discriminated from background noise, clearly and with high probability close to unity. To this end, the authors use, as input microwave photon, single microwave photons (or weak coherent-state pulses, which are composed only of the zero and one-photon states) and define the detection efficiency by the ratio of "counts" and the input photon numbers.

In the current form, this manuscript only reports efficient conversion of microwave photons into electric current. The reported value is rather low in comparison with the prior works using superconducting qubits. For these reasons, I do not recommend publication of this manuscript in Nature Communications.

Comments:

(1) Most previous works on single microwave photon detector use superconducting atoms as receivers of microwave photons. I would like to ask the authors to summarize what are the advantages/disadvantages of the detectors using semiconductor quantum dots.

(2) After the dot absorbed a single photon, it transits to the excited state and cannot absorb the next incoming photon. Therefore, it seems that there exists a dead time after detection. If it is correct, what is the order of dead time? Are there any methods to reset the dot to the ground state immediately?

(3) The authors say that the efficiency is 7% in Figure 2 and 6% in the main text. They should briefly explain this discrepancy.

Reviewer #3:

Remarks to the Author:

The manuscript "Efficient and Continuous Microwave Photodetection in Hybrid Cavity-Semiconductor Nanowire Double Quantum Dot Diodes" presents an experimental realization of a potential microwave single photon detector based on an hybrid device coupling a superconducting coplanar waveguide resonator to a semiconductor double quantum dot (DQD) charge system. The system confining the electrons is realized in a crystal phase defined DQD embedded in InAs nanowires.

Recently considerable attention has been devoted to hybrid semiconductor-superconducting systems, representing a very interesting platform to study both fundamental physics and spin/charge-based qubits coupled to microwave photons.

InAs nanowires have also been extensively studied in the past, motivated largely by their strong spin-orbit interaction allowing all-electrical control of spin bits. Despite a series of initial interesting attempts, progress in the field originally proceeded quite slowly given the difficulties in combining high-quality superconducting resonators to the tiny electric dipole moment of semiconductor DQDs, which on the other side usually presents a very high decoherence rate.

The situation has changed drastically in the last few years thanks to a number of groundbreaking experiments, demonstrating the strong coupling regime between both the charge and spin degrees of freedom of electrons confined in GaAs and SiGe double and triple quantum dots and photons confined in standard 50 Ohms and high impedance superconducting resonators.

The work presented in this manuscript represents an experimental step toward the use of this class of hybrid devices as a potentially efficient single microwave photon detector. This proposal has been originally presented in Ref{4}, correctly cited by the authors.

Despite representing a nice experiment, I find the work presented in this manuscript non-convincing in its current version and partially just incremental respect to what already presented by other similar experiments; therefore, I do not support its publication in Nature Communications in its current version. I will be willing to reconsider it if the authors manage to address my concerns here below.

Similar devices have been reported and extensively studied in Petta's group in Princeton and Wallraff's group in ETH. From the device point of view, the main element of novelty is represented by the DQD defined in InAs zinc-blend-wurtzite heterostructure, previously presented by the same authors in Ref{11}. Similar measurements of DQD photocurrents have been reported in Refs{18,25}.

The main measurements of the current experiment, reported in Figure 3, is not well explained. In a photon assistant transport process (PAT), one expects to observe current in the DQD charge stability diagram in a region a finite detuning, parallel to the DQD interdot region (zero detuning) where the resonant condition is matched. But the current distribution observed in Fig.3(a) does not follow this typical expected PAT pattern. It is instead a broad continuum of current (also out of resonance), not just along the dashed region depicted by the authors for positive and negative detuning. Do the authors understand this behavior or are able to model it? If so, adding this to the manuscript could help the reader understanding the process.

Furthermore, I have the following concerns:

1) It is not clear in the manuscript if the authors are able to estimate the charge decoherence rate for electrons confined in the DQD. On page 3 the authors write: 'The effect of the DQD is captured by a change in linewidth given by an additional loss term $\kappa_{\text{DQD}} = 4g^2/\Gamma$ as an additional channel opens up for the photons to escape the resonator', but it is not immediately clear what ' Γ ' represents. As indicated in the original theoretical proposal in Ref.{4}, this term should also contain a contribution coming from the intrinsic decoherence of the DQD (see for example Eq.18 of the cited work). For a sufficient fast dot-lead tunnel rate one could neglect the intrinsic DQD decoherence. The DQD used in this work presents quite high charging energy and it is used a finite detuning (far away from the sweet spot). Therefore, it should present a substantial decoherence rate, usually higher than a GHz. Instead, the typical tunneling rates reported in this manuscript are of few tens of MHz. This is an important point that would be worth clarifying in the manuscript.

2) At the end of the manuscript the authors write: 'Moreover, the demonstrated efficiency is two orders of magnitude higher than the one for photoemission in similar hybrid cavity structures [18, 25]. The reason is that the key mechanism suppressing the photoemission efficiency, DQD relaxation via spurious emission of phonons, is too slow to affect the photodetection process which relies on fast electron tunneling out of the excited DQD.' This sentence should be supported by measurements. Did the author estimate the relaxation and dephasing rates for their DQD tuned in the studied configuration? Usually, at finite DQD detuning the relaxation rate gets suppressed and the main source of decoherence is represented by $1/f$ charge noise-induced dephasing.

3) It will be very informative and important for the manuscript to provide an estimate of the relaxation and dephasing rates for the DQD charge system and to compare those rates with the electron tunneling rates across the DQD (Γ in the manuscript).

4) I do not find trivial the sentence 'The second and the third condition have a trade-off because $g \propto t$ '. Could the authors briefly describe (in the manuscript) why $g \propto t$?

5) The resonance condition can be realized for different interdot tunneling rate (t) and for different DQD-detunings. How this impacts the quantum efficiency of the photon detection process?

6) The estimated coupling strength $g=3.6$ MHz is quite low. I guess this is the coupling strength at some finite detuning (renormalized by the mixing angle). It would be useful to quote also the bare

coupling strength (estimated for zero DQD-detuning) in order to directly compare it with other studied similar hybrid devices. Do the authors think that the detection efficiency would benefit by promoting the system in the strong coupling regime?

7) It is not completely easy to follow the reasoning of the authors in explain how they estimated the microwave power at the detector. For example, it seems that they assume 'identical' lines (the 3 lines reported in Fig.1 of suppl. Material). With this assumption and by measuring the signal power reflected for a signal frequency well detuned from the resonator, they try to estimate the extra attenuation introduced by the cable. This assumption is not true in general. The transmission input and RF output are not identical (there is the amplifier in the output line, so at least 2 more SMA connectors). In general, even if the 2 lines would have built nominally identical, there are always impedance mismatches introduced by connectors and by the different rf components (for example usually a cryogenic amplifier presents a VSWR not =1). Furthermore, this attenuation can also be frequency-dependent. Therefore, estimating the attenuation far detuned from the resonator fundamental mode can introduce some miscalibration. In addition, from the transmission line, there is another set of bonding wires (to the left port of the resonator) that the microwave photons do not probe while sent through the RF reflection (R) port.

Dear Editor, dear Referees,

We thank the Referees for the in-depth review of our manuscript, providing constructive comments and criticism. To address the comments of the Referees we have performed additional experimental data analysis and extended numerical calculations. We have also substantially revised our manuscript, both the main text and the supplemental material. The revision has however not changed our main results. To provide an accessible, albeit detailed response, we first comment on the overall assessment of the three Referees. Below we then answer, point-by-point, to all the comments and criticism of each Referee.

The first Referee finds our results to be *“of interest for a broad audience”*. Their main concern, regarding the soundness of our claims, is related to the question of which photons our detector detects: photons stored in the cavity or itinerant photons in the transmission line. As we clarify in the updated manuscript, the cavity should be understood as part of the detector. Just like a conventional photodiode, our detector detects the photons with which it is illuminated. In our setup, these are the itinerant photons in a transmission line. In addition, the Referee raises several points that deserve clarifications. We have addressed all these points, providing the detailed information requested by the Referee and further improving the readability of the manuscript.

The second referee does not recommend publication because they find our *“claim that this is the first continuous microwave photodetection [...] too strong”* and because *“the reported value is rather low in comparison with the prior works using superconducting qubits”*. The first of these statements appears to be based on the perception that a photodetector requires the capability of detecting single photons, which is not true for most conventional photodetectors. In our work, as the Referee correctly points out, we report *“efficient conversion of microwave photons into electric current”* in complete agreement with our claims in the abstract. Concerning the comparison with superconducting qubits, we stress that our architecture has the key advantage of not requiring any detection timing with respect to incoming photons. Our results thus open a novel avenue that complements previous photodetectors based on superconducting qubits, and we are convinced that much higher efficiencies are attainable once our approach reaches a similar maturity.

The third referee does not recommend publication in its current form but is *“willing to reconsider it if the authors manage to address my concerns”*. In particular, the Referee is concerned about the novelty of our results, pointing to previous literature. While similar cavity-double dot architectures have been investigated before, to the best of our knowledge there are no experiments where high-efficiency detection of microwave photons have been reported. Indeed, the low efficiencies observed in cavity-double dot photoemission experiments rather imply that efficient photodetection is challenging, if not impossible. For this reason, we believe that our results considerably go beyond the state-of-the-art, with a large potential impact on future technology. The Referee further raises concerns regarding the explanation of the experiments, in particular our treatment of decoherence and the microwave power calibration. In response to the comments of the Referee, we provide a detailed information about the calibration procedure and measurements. We also updated the analysis and substantially revised the manuscript, improving readability and the connections to previous results. In particular, the role of decoherence is now fully addressed in our revised theoretical modelling.

To summarize, we have comprehensively addressed the Referees' concerns and performed an appropriate revision of the manuscript. In our opinion, the manuscript as it now stands fulfills the criteria for publication in Nature Communications.

On behalf of all the authors,

Ville Maisi

Response to Referee 1 (black: Referee comments, blue: our replies)

In their manuscript, the authors claim the efficient and continuous detection of microwave photons demonstrating the functionality of a photon detector implemented in a nanowire double quantum dot (DQD) diode coupled to a superconducting coplanar resonator. They fabricated a crystal phase defined nanowire DQDs in correspondence of the electric antinode of the resonator and measured the source-drain current which, in resonant conditions, is attributed to a photon-assisted tunneling process between DQD levels. By measuring magnitude and phase of reflection and transmission coefficients of the resonator they also obtain additional information on level spacing and photon absorption in the DQD detector.

Microwave photon detectors based on DQDs were reported a long time ago but, as the authors correctly pointed out in their manuscript, the efficiency of such detectors was very low. In recent times, there has been a growing interest in the development of suitable methods for the detection of single microwave photons. While important results have been obtained with superconducting circuits, semiconductor DQD-based detectors have lagged behind. The working principle of the latter is basically photon-assisted tunneling in DQD devices, which is not new for its own sake, nor is the hybrid DQD-resonator architecture. However, experiments that define conditions for developing efficient microwave photon detectors deserve attention. This manuscript certainly covers a very interesting topic and I think it could be of interest for a broad audience that extends beyond the already large communities of semiconductor devices and hybrid quantum systems.

There are however a series of points in which the reported results are not convincing, thus I have serious concerns regarding the soundness of their claims. Several experimental details are missing and the readability of the manuscript should also be improved. In the following I list my comments and questions, which must be addressed before I can recommend the manuscript for publication.

1. In the abstract, the authors claim the efficient and continuous detection of “itinerant” photons by the nanowire double quantum dot. Strictly speaking this is not correct: in their experiment, the detector probes the photons stored in a cavity, not directly the itinerant photons entering from the transmission line. This point has been stressed in different experiments with superconducting circuits [see e.g. *Nature Physics* 6, 663 (2010), *Nature Physics* 14, 546 (2018) and *Phys. Rev. X* 8, 021003 (2018)]. The authors should correct their claim.

Reply: The question raised by the referee if the photons arrive to the detector from outside or not is very important: a detector that detects photons arriving to it is much more useful than one that detects photons inside the system. We are, however, puzzled by the claim of the referee that in our experiment, “strictly speaking” itinerant photons are not detected, instead “the detector probes the photons stored in the cavity”. To support the claim the referee points to our Ref. [29] “Quantum non-demolition detection of an itinerant microwave photon”, *Nature Physics* 14, 546 (2018), where itinerant photons in a transmission line are detected by a superconducting qubit embedded in a microwave cavity. However, in a qualitatively almost identical fashion (but clearly not QND), in our experiment itinerant photons in the transmission line are detected by a double quantum dot embedded in the cavity.

It is here important to emphasize that our detector consists of the combined cavity-double quantum dot system. Consequently, the efficiency of the detector is defined as the number of photons detected divided by the number of photons incident on the detector, or equivalently, as the conversion factor between the measured electrical current and the influx of photons, see our Eq. (1). However, the microscopic processes inside the detector do neither affect the definition of the efficiency, nor does it

change the fact that it is itinerant photons that are detected. To avoid misunderstandings on this point we have modified our schematic of the experimental set-up in fig. 1a and also the text in the caption. We also extended and reformulated the paragraph defining the efficiency.

2. In Fig. 2 photocurrent data are presented as a function of the microwave power. How many photons are on average in the resonator? Could the authors evaluate the conversion factor between the measured current and the photon number?

Reply: The average number of photons in the cavity is approximately 15 for a measured current of 2.4 pA and photon power 1 fW, with the system parameters given in Fig. 2. Dividing the measured current by the number of photons thus gives 0.16 pA per photon in the cavity. To provide the reader with this additional information about our experiment, we have added a discussion together with these estimates to the supplemental material.

We however emphasize that neither the average number of photons in the cavity nor the ratio between the measured current and the photon number is, in our opinion, central to the photo-detection scheme in our experiment. As mentioned in the previous response point, the key quantity is the detection efficiency, defined as the conversion factor between the measured current and the influx of photons. We are thus interested in the *rate* at which photons impinge on the detector and the *rate* by which the electrons are transferred through the double quantum dot. The average number of photons does not provide direct information about these rates. As a concrete example, by simply increasing all energy scales by a factor of two, the number of photons in the cavity would be unchanged while the output current would be doubled, providing 0.32 pA per photon. In order not to distract the reader from the photo-detection scheme described in the manuscript, we thus refrain from discussing the photon number and the current per photon in the main text and instead present this in the supplemental material.

3. For the data in Figure 2, bias and gate voltages are not indicated, thus the experimental conditions are not clear. In addition, details on the theoretical curves are not given, the reader has to scroll down to the end of the paper in the Methods section in order to find information on the fit. A more detailed description of Figure 2 would be appreciated.

Reply: To specify the operation conditions, we modified the caption of Fig. 2 which now includes *“The incident microwave drive signal is in resonance with the resonator, $f = f_r$, and the DQD, $E = hf$, at a detuning $\delta = + \delta_r$ (c.f. Fig. 3). No bias voltage is applied.”*

Furthermore, to give the reader information on the theory curves early on in the paper we added text below Eq. (1) *“The experiment is modelled theoretically within a framework based on the Jaynes-Cummings Hamiltonian (see Methods). We find excellent agreement between experimental data and theory curves, see Figs. 2 and 3”.*

Taken together we think these additions make a more detailed description of Fig. 2 superfluous.

4. In my opinion, it is odd to show the Fig. 2 (photoresponse and quantum efficiency as a function of power) before Fig. 3 (electronic and photonic response as a function of gate voltages and frequency). I think the other way round would be clearer.

Reply: We see the point of the Referee, however, targeting a broad audience we prefer to present an overview of the main results, which includes Fig. 2, in the beginning of the paper. The more detailed, technical parts, including Fig. 3, are presented later in the paper (and in Methods and Supplement).

We feel that this style of presentation is in line with the policy of the Nature journals, to make the results accessible to non-experts.

5. Page 2, left column. The authors state that a photocurrent is observed when the energy gap of the double quantum dot (DQD) matches the photon energy. The numerical values of inter-dot tunneling coupling (t) and detuning (δ_r) are given only in the supplemental material. I recommend indicating these numbers also in the manuscript in order to strengthen the analysis of the experimental results.

Reply: As recommended by the referee, we have included the values of tunnel coupling and detuning δ_r in the main text.

6. In Methods section the authors state that “Shifts in the voltages V_{LP} and V_{RP} arising from offset charge changes were compensated in the course of the measurement to keep the photodetector in the same operation point.” This behavior suggests that the charge configuration of the nanowire DQD is not stable. I wonder how the authors can safely assert that the operation point is the same after the reconfiguration of the DQD. Could the authors clarify this point?

Reply: We are confident that we did all the measurements for the same charge configuration. To safely assert this, we did the following: Whenever a larger change in the gate voltages V_{LP} and V_{RP} occurred, we measured the finite bias triangles around the charge configuration. Each charge configuration has a pair of bias triangles with distinct features, acting like a “fingerprint” (characterized by the magnitude and functional dependence of the current within the triangle), at $V_b = 1\text{mV}$. In addition, we repeated the photo-current measurement of Fig. 3a of the main manuscript to assure that the photodetection response is still the same.

To make it evident to the reader that all measurements are done for the same charge configuration, we added a new paragraph “*Charge configuration*” to the supplemental material. The paragraph includes a new figure (Fig. 3) which presents the control measurements. In the figure we summarize three sets of such measurements to illustrate how the drift in gate voltages occurred.

7. The stability diagram is shown only as supplemental material and for a range of plunger gate voltages different from those in Fig. 3a and b. The configuration of the double quantum dot for the data in Fig. 3 is not clear. What is the source-drain bias voltage? Is there an estimation of the electron occupancy? Fig. 3a and 3b have different values of the plunger gate voltages. Direct comparison of data taken in different conditions is misleading, the authors should explain why measurements are not taken in the same conditions. What is the voltage of the plunger gates in Fig. 3c and d?

Reply: The charge stability diagram is shown only in the Supplementary since this is a standard tool for characterizing the DQD properties and not central to the main story. As pointed out in the response to comment 6, there is a drift in V_{LP} & V_{RP} during the measurements, explaining that the values are different from those in Fig. 3a and b. The drift also explains why Fig. 3a and 3b have different values of the plunger gate voltages. We however stress again that we carried out extensive control measurements (see added paragraph “charge configuration” in Supplement) to make sure that our measurement is at the same charge configuration.

To clarify the configuration of the DQD for the data in Fig. 3 we added the values for the plunger gate voltages to the caption for Fig. 3c and d. We also state in the caption that there is no source-drain bias voltage applied.

The electron occupancy, or the number of electrons in the DQD, cannot be estimated, since the electrical current becomes vanishingly small before reaching the last electron. We stress, however,

that the DQD electron number is not relevant for the obtained results, what is crucial is instead that we can address individually one energy level per dot (which we demonstrate). Therefore, the electron occupancy is not discussed in connection to Fig. 3.

8. In Fig. 3a, the photocurrent peaks are relatively large spots along the detuning axis. What are the factors that determine such a large broadening? Why there is a measurable photocurrent when resonator and DQD are not resonant ($\delta \neq \pm \delta_r$)? Additionally, in Fig. 3a the reversal of the current polarity is visible for δ near zero. Why?

Reply: The large spot photocurrent peaks in Fig. 3a are simply due to resonance broadening. The broadening has different physical origins: photon loss, dephasing, coupling to source and drain, and phonon relaxation. To clearly explain the origin of the broad peaks we added text in connection to Fig. 3 as “The extension of the photocurrent peaks perpendicular to the constant detuning line are due to resonance broadening and agree well with the theoretical low power prediction of $\tilde{\Gamma} + 4g^2/\kappa$, where $\tilde{\Gamma}$ denotes the decoherence rate, g the coupling strength of the DQD to the resonator, and κ the resonator line-width.” To illustrate the theoretical prediction for the resonance broadening, we added lines to Fig. 3a.

The reversal of the current polarity for δ near zero (most likely) arises from a rectification effect similar to what is presented in Cornia et al., Scientific Reports 9, 19523 (2019) for single quantum dot. At zero detuning the DQD levels hybridize fully, spreading over both dots equally and making the energy levels effectively similar to the levels of a single quantum dot. To better explain the polarity reversal to the reader, we added the reference Cornia et al. and a comment to the Supplementary as “We anticipate that the current reversing response at $\delta = 0$ of Fig. 3a in the main article arises from similar effect as reported in Ref. 7”. In this context it is also worth noting that the earlier DQD-microwave experiments [see for example Rev. Mod. Phys. 75, (2003) and references therein] also showed a response at zero detuning.

9. I am puzzled by Fig. 3a. The photon-assisted tunneling process typically gives rise to current peaks when the spacing between left and right dot levels matches the microwave frequency. Once that the detuning is fixed, the resonance condition can be satisfied also by slightly shifting the levels of both dots with respect to the leads. This is what is observed in the phase measurement in Fig. 3b. Accordingly, I would expect that the photocurrent forms two “ridges” aligned along the normal of the detuning axis: this is what is typically observed in microwave spectroscopy of DQDs [see e.g. Rev. Mod. Phys. 75, (2003), pages 14-17]. However, Fig. 3a is in stark contrast with this behavior: nonzero current is observed along the detuning axis, while in the normal direction the current is null. Why? Does it mean that the spacing between DQD levels and the leads matters, thus there is an effect of the leads? I think this is a crucial point, the authors should explain this discrepancy between photocurrent, phase data and the literature. A schematic diagram of the DQD levels would be useful in order to explain the tunneling process related to Fig. 3a.

Reply: The referee points out that the photocurrent is expected to form “two ridges aligned along the normal of the detuning axis”. This is typically seen in microwave spectroscopy of DQDs and illustrated schematically in e.g. Fig. 21 in Rev. Mod. Phys. 75, (2003), mentioned by the referee. Based on this observation, the referee claims our result in fig 3a is “in stark contrast” with what is expected, showing that “nonzero current is observed along the detuning axis, while in the normal direction the current is null”.

Importantly, in contrast to what is claimed by the referee, our result in Fig. 3a is in perfect agreement with expectations as well as with the literature. In fact, we have performed additional analysis of the measurement data and confirmed that the length and width (in energy) of the ridges is given by $h\nu$.

and $\sqrt{\Gamma} + 4g^2/\kappa$ respectively, as predicted by theory (see also discussion below and in the previous response point). We stress that the form of the ridges in the charge stability diagram is affected by the lever arms of the plunger gates. In our experiment, this results in ridges in Fig. 3a which are wide (elongated along the detuning axis) and short (compressed along the normal to the detuning axis).

To clarify this important point and to avoid giving the reader the impression that our photocurrent results are in conflict with the existing literature, we have modified Fig. 3a (now displaying energy scales for the length and width of the ridges) and also updated the caption of Fig. 3. and the corresponding manuscript text.

Let us also comment on the question by the referee “Does it mean that the spacing between DQD levels and the leads matters, thus there is an effect of the leads?”. The answer is yes, the energy levels of the DQD with respect to the leads matters, as is also clear from the discussion in Rev. Mod. Phys. 75, (2003). For the photodetection operation an electron needs to be present in the DQD ground state, i.e., the ground state needs to be below the chemical potential of the leads. At the same time, the excited state needs to be above the chemical potential such that the electron can leave the DQD upon absorbing a photon. If one of these conditions is not met, the photo-current drops to zero. It is in fact these conditions that determine the length of the ridge to be $h\nu$. Since this is a well-known property of the photocurrent, we feel that a “schematic diagram of the DQD levels”, as proposed by the Referee, is superfluous.

10. Data in Fig. 3a are taken with microwave power of 2 fW, control data taken without external microwave drive are not shown. Is there a background current? Could it be used to quantify the dark count rate of the detector?

Reply: We have measured the photocurrent without external microwave drive. There is no measurable signal at $\delta = \pm \delta$, however, a miniscule signal is observed at $\delta = 0$. This allows us to conclude that the background current is below the noise level and hence, that we cannot use this measurement to quantify the dark count rate of the detector. To explain this to the reader, we have added a figure which shows control data of the photodetector without microwave drive, to the Supplementary (Fig. 2). In connection to the figure, we also added the text “*Fig. 2 presents the photodetector response with no applied microwave signal. We observe a miniscule signal at the charge triple point at $\delta = 0$. At the photodetector points $\delta = \pm\delta$, we see no measurable current. Thus, the dark current of the photodetector is below the 0.2 pA noise level.*”

11. Could the authors exclude the presence of excited states within the bias window? There seems to be a non-constant current in the bias triangles shown in Fig. 1a of supplemental material.

Reply: We thank the referee for raising this point. The presence of excited states within the bias window can indeed be excluded based on our measurement results. To demonstrate how this is done, providing also an explanation for the non-constant current in the bias triangles, we added the following text to the Supplementary: “*In Fig. 1a, we see the first excited state with pronounced current in the middle of the triangles. The bias voltage opens up an energy window of 400 ueV. As the excited state is in the middle of it, we estimate the higher excited states of the quantum dots to be approximately 200 ueV above the states considered for the photodetector. Since this energy is an order of magnitude larger than the photon energy, and other energies in the system such as the thermal energy $k_B T$, the excited states of the quantum dots do not influence the photodetector operation.*”

12. On page 2, while commenting on Fig. 3b the authors state that “The dispersive phase shift of the resonator transmission displays also the two resonances $\delta = \pm \delta_r$ ”. This sentence is a little

clumsy and needs to be reformulated. The dispersive limit takes place for large detunings, while for $\delta = \pm \delta_r$ the resonator and the DQD are on resonance.

Reply: We agree with the referee that the sentence is not optimally formulated, the term “dispersive” is not appropriate here. We therefore reformulated the sentence such that the word “dispersive” is no longer used.

13. On page 2, while commenting on Fig. 3c and d, the authors state that “the photocurrent follows the same Lorentzian lineshape as the resonator response”. The fitted values of resonance frequency and linewidth are not reported. Could the authors be more specific on this point? What is the frequency selectivity, hence the bandwidth, of the detector?

Reply: The referee asks us to be more specific and to provide fitted parameter values. A careful analysis gives the resonance frequency $f_r = 6.436$ GHz and linewidth of 15.5 MHz, within the measurement accuracy the same as for the resonator transmission and reflection. The detector bandwidth is just given by the linewidth. To provide the reader with this information and to clarify how the values have been obtained, we modified and extended the passage mentioned by the referee to “At $\delta = \pm \delta$, the photocurrent lineshape, for both positive and negative current peaks, is within measurement accuracy the same as the one of the resonator response in Fig. 3c. We note that the detector bandwidth is given by the resonance linewidth.” We also added a short description in Methods, about how the linewidths were determined and what are the values for each resonance peak and dip.

14. On Page 3 it is stated that “We observe the transmission decreasing by $\Delta T = -0.06$ and the reflection increasing by $\Delta R = 0.02$ ”. This variation is relatively small, how wide are the error bar?

Reply: The errors were estimated to “ ± 0.01 ” for both the ΔT and ΔR values, based on the scatter of the experimental data points. We added the error estimates to the sentence.

15. On Page 3 below in the same paragraph it is stated that “This reduction exemplifies that the DQD absorption removes photons from the cavity and it is comparable to but interestingly smaller than η .” It would be useful to check this behavior also at different microwave power, thus different efficiency.

Reply: We agree with the referee that it would be interesting to check this behavior at different efficiencies, it would provide additional information and serve as an extra test of how well the theory can explain the experimental data. However, the main results of the paper do not depend on the outcome of such a check. We therefore refrain from performing such new measurements within our present work and instead leave this to a future, follow-up study.

In this context, let us point out that we see a number of equally interesting topics for a follow-up study. For example, the dependence of the photon losses and efficiency on the directivity p_r at different inter-dot tunnel couplings, on the source and drain reservoir tunneling rates (including asymmetry), cavity resonance frequency dependence as well as on the bias voltage.

16. On page 3, left column, the authors state that “The effect of the DQD is captured by a change in linewidth given by an additional loss term $\kappa_{\text{DQD}} = 4g^2/\Gamma$ ”. However, g and Γ are not defined at this point. In the same paragraph, the detection is said to be “switched on”. Do the authors mean that the double quantum dot is polarized with detuning equal to δ_r ?

Reply: We thank the referee for pointing out the missing definitions of g and Γ when they are introduced in the text. We have reformulated the text such that g and Γ ($\tilde{\Gamma}$ in the updated manuscript) are defined when they are introduced.

“Switching on” refers to tuning the DQD away from Coulomb blockade to the photodetection point. We have added text to the paragraph to clarify this.

17. The total decay rate of the resonator is $\kappa = \kappa_L + \kappa_R + \kappa_{\text{int}} = 15.5$ MHz, it thus follows that the loaded quality factor is $Q = f_r / \kappa = 415$. This value is significantly lower than what reported in the literature. Large κ_L and κ_R follows as a consequence of the large capacitances between resonator and feed lines. However, the internal loss rate is quite high, even with LC filters included in the dc lines to suppress microwave leakage. Do the authors have an explanation for this behavior?

Reply: We agree that the value of the loaded quality factor is lower than what is reported in the literature. Measurements on additional resonators, performed after the present study, suggest that the silicon substrates that we used have excessive losses. We are currently working on minimizing the losses and obtaining higher quality factor resonators, in line with what other groups have achieved experimentally. The origin of the internal losses is however not critical for the validity of our results. Therefore, we refrain from speculating about possible origins in the manuscript.

18. Page 4, left column. “Based on previous work on microwave resonators interacting with quantum dots [17, 19, 21]...” Several references are missing, see Sensors 20, 4010 (2020).

Reply: We agree with the referee that we here do not refer to all the works on microwave resonators interacting with quantum dots. This was also not our intention. With the chosen references we wanted to point to experiments with high quality factor resonators interacting with double quantum dots. To make this more apparent we moved the citations to a more appropriate place in the sentence. It now reads *“Based on previous work on microwave resonators interacting with quantum dots, such high quality factor resonators and larger source-drain tunnel rates $\Gamma / 2\pi \sim 1$ GHz are experimentally obtainable [17, 19, 21] by adjusting the geometry of the input coupling capacitance and minimizing the internal losses of the resonator. This thus makes near-unity quantum efficiency detectors experimentally accessible.”* If the referee think we still omitted any relevant references, we are of course willing to add them.

Let us also thank the referee for pointing out the the review article Sensors 20, 4010 (2020). It is a very recent, excellent review of the possibilities to perform photodetection with DQDs. We unfortunately oversaw this review during the preparation of our paper but have now added it to the introductory part of our manuscript, where we present an overview of the field.

19. Page 4, left column. The coupling strength is said to be proportional to the tunneling rate. A reference would be useful here.

Reply: We added Ref. 22 (Childress et al. 2004) here, where this dependence is reported.

20. In ref. Phys. Rev. A 95, 012325 (2017), the efficiency at the optimal point is obtained by tuning inter-dot tunnel coupling, DQD-resonator coupling strength and total decay rate of the resonator. The tunneling rate between dots and leads also plays a role. The efficiency determined by Eq. 2 does not seem to be directly dependent on such parameters. Could the authors better explain these differences with the model by Wong and Vavilov?

Reply: We stress that there is no difference between our result in Eq. (2) and what is presented by Wong and Vavilov, Phys. Rev. A 95, 012325 (2017). In fact, our Eq. (2) depends on all these parameters mentioned by the referee, in the same way as described by Wang and Vavilov. To see this, note that $\kappa_{\text{DQD}} = 4g^2 / (\Gamma + \Gamma_r + 4\Gamma_\varphi)$, as described on page 3, where we have the DQD-resonator coupling

strength g and the tunneling rate Γ between dots and leads. The total decay rate of the resonator ($\kappa + \kappa_{\text{DQD}}$) is already explicitly visible in Eq. (2). Furthermore, g in the above equation as well as the directivity p_r in Eq. (2) depend on the inter-dot tunnel coupling t . With these relations we obtain all the dependencies that the referee pointed out.

To emphasize that our Eq. (2) is perfectly consistent with the results of Wong and Vavilov, we modified the sentence before Eq. (2) to contain “*in line with Ref. 4,*”. Please note also the updates made to include the DQD relaxation Γ_r and dephasing Γ_ϕ in the analysis. These updates are discussed in the response to questions 1 and 3 of Reviewer #3.

Response to Referee 2 (*black: Referee comments, blue: our replies*)

In this work, the authors demonstrated high-efficiency conversion of microwave photons into an electrical current, using a double quantum dot as an absorber of single microwave photon. From the slope of the current-microwave power plot, they concluded the efficiency of conversion reaches 7%. The electronic and photonic responses (Figure 3) support this estimation. From these data, the authors argue that this is the first realization of continuous photodetection in the microwave domain.

However, in my opinion, the authors’ claim that this is the first continuous microwave photodetection is too strong. The authors should also demonstrate that the electric current induced by a single microwave photon can be discriminated from background noise, clearly and with high probability close to unity. To this end, the authors use, as input microwave photon, single microwave photons (or weak coherent-state pulses, which are composed only of the zero and one-photon states) and define the detection efficiency by the ratio of “counts” and the input photon numbers.

Reply: The Referee finds our claim of continuous microwave photodetection too strong, seemingly because we do not demonstrate detection of single photons. We disagree with the referee that a photodetector must have the capability of detecting single photons: there are many examples of conventional photodetectors which do not have such capabilities (e.g., phototubes and CCDs). A prominent example is the conventional photodiode to which we compare our device in the manuscript. The requirement of the Referee is thus not in agreement with the standard terminology in the field and we therefore stand behind our claim that we experimentally implemented a continuous microwave photodetector.

In the current form, this manuscript only reports efficient conversion of microwave photons into electric current.

Reply: In line with our previous reply point, to convert a continuous stream of incoming photons to an electrical current is the conventional principle of operation for a photodetector.

The reported value is rather low in comparison with the prior works using superconducting qubits.

Reply: We agree with the Referee that our detection efficiency is lower than in works using superconducting qubits. However, this is already pointed out in the manuscript and is not of key importance for the significance or novelty of our results, see next response point.

For these reasons, I do not recommend publication of this manuscript in Nature Communications.

Comments:

(1) Most previous works on single microwave photon detector use superconducting atoms as receivers of microwave photons. I would like to ask the authors to summarize what are the advantages/disadvantages of the detectors using semiconductor quantum dots.

Reply: The main advantage of our semiconductor quantum dot-based detector is that it is continuous in time, while superconducting qubit-based detectors can only detect photons arriving at predetermined times. In addition, while a detected photon in our system directly gives rise to a transferred electron (that is, a current), for qubit-based detectors one typically has to apply a sequence of readout pulses to the qubit to determine if the photon was detected or not. In the manuscript these advantages are already emphasized: *“In addition and of key importance for photocounting applications, the photodiode operation is continuous in time; a detected photon directly gives rise to an electron transfer through the DQD. This is in stark contrast to state-of-the-art single microwave photon detectors based on superconducting qubits [10, 26–29], where photodetection, despite of reaching close to unity efficiency and demonstrating non-demolition capability, is indirect, occurring via qubit readout only at predetermined times.”*

The present disadvantages, predominantly the lower efficiency and the absence of single photon detection, are in our opinion mainly due to the different maturity of the two schemes. While superconducting qubit-based detectors have been analyzed experimentally during at least half a decade, our work is the first demonstration of a semiconductor quantum dot based photodetector where the efficiency could be determined (and, arguably, was sizeable). As discussed in the manuscript, we see no fundamental obstacles for performing single-photon detection with close-to-unity efficiency, in fact in forthcoming work we will be aiming for achieving these features.

(2) After the dot absorbed a single photon, it transits to the excited state and cannot absorb the next incoming photon. Therefore, it seems that there exists a dead time after detection. If it is correct, what is the order of dead time? Are there any methods to reset the dot to the ground state immediately?

Reply: The referee is correct, there is a dead time after detection. As we discuss in the updated manuscript and in detail in the response to Referee 3 below, there are two dominating processes leading to a resetting of the DQD to its ground state: i) Electron tunneling out from the from the DQD (excited state) to the drain followed by another electron tunneling in to the DQD (ground state) from the source. This occurs on a time scale $1/\Gamma$, ii) Relaxation of the electron from the excited to the ground state of the DQD, via emission of a phonon. This occurs on the time scale of $1/\Gamma_r$. The dead time is hence given by $1/(\Gamma + \Gamma_r)$, estimated to be 14 ns for our device.

To obtain a high efficiency detector, the rate Γ_r should be minimal. Therefore the $1/\Gamma$ is the relevant quantity limiting the dead time. We now provide this deadtime $1/\Gamma = 20$ ns in the manuscript. To our mind this is the fairest number as it does not take into account the speedup from the spurious relaxation process. To provide this information to the reader, we have added text to the manuscript *“This tunnel rate also characterizes the dead time of the detector $\sim\Gamma^{-1} = 20$ ns.”*

Concerning an immediate/fast reset to the ground state, the tunneling rate Γ can be increased. Values of the order $\Gamma = 1$ GHz are achievable in DQD systems experimentally. This would yield a shorter dead time of 1 ns. However, this would have a trade-off with the detector bandwidth as κ_{DQD} depends on Γ and for high efficiency one needs the resonator linewidth $\kappa = \kappa_{\text{DQD}}$.

(3) The authors say that the efficiency is 7% in Figure 2 and 6% in the main text. They should briefly explain this discrepancy.

Reply: The 7 % efficiency is the low power, linear response theory result while the 6 % is what we realized in the experiment. We see from the comment of the referee that giving these two different numbers at different places in the manuscript can create confusion. We have therefore removed the 7% text from Fig. 2 and also modified the caption to Fig. 2 and the manuscript text at appropriate places. We note that including decoherence and dephasing now result in a theoretically predicted low-power limit of 6.6%.

Response to Referee 3 (*black: Referee comments, blue: our replies*)

The manuscript "Efficient and Continuous Microwave Photodetection in Hybrid Cavity-Semiconductor Nanowire Double Quantum Dot Diodes" presents an experimental realization of a potential microwave single photon detector based on an hybrid device coupling a superconducting coplanar waveguide resonator to a semiconductor double quantum dot (DQD) charge system. The system confining the electrons is realized in a crystal phase defined DQD embedded in InAs nanowires.

Recently considerable attention has been devoted to hybrid semiconductor-superconducting systems, representing a very interesting platform to study both fundamental physics and spin/charge-based qubits coupled to microwave photons.

InAs nanowires have also been extensively studied in the past, motivated largely by their strong spin-orbit interaction allowing all-electrical control of spin bits. Despite a series of initial interesting attempts, progress in the field originally proceeded quite slowly given the difficulties in combining high-quality superconducting resonators to the tiny electric dipole moment of semiconductor DQDs, which on the other side usually presents a very high decoherence rate.

The situation has changed drastically in the last few years thanks to a number of groundbreaking experiments, demonstrating the strong coupling regime between both the charge and spin degrees of freedom of electrons confined in GaAs and SiGe double and triple quantum dots and photons confined in standard 50 Ohms and high impedance superconducting resonators.

The work presented in this manuscript represents an experimental step toward the use of this class of hybrid devices as a potentially efficient single microwave photon detector. This proposal has been originally presented in Ref{4}, correctly cited by the authors.

Despite representing a nice experiment, I find the work presented in this manuscript non-convincing in its current version and partially just incremental respect to what already presented by other similar experiments; therefore, I do not support its publication in Nature Communications in its current version. I will be willing to reconsider it if the authors manage to address my concerns here below.

Similar devices have been reported and extensively studied in Petta's group in Princeton and Wallraff's group in ETH. From the device point of view, the main element of novelty is represented by the DQD defined in InAs zinc-blend-wurtzite heterostructure, previously presented by the same authors in Ref{11}. Similar measurements of DQD photocurrents have been reported in Refs{18,25}.

Reply: It is indeed true that DQD-resonator structures like ours have been reported by other groups (as also discussed in our manuscript). Moreover, as the Referee says, we use DQDs defined with a crystal phase InAs nanowire. These structures yield textbook like features in the experiments and thus our view is that they play a key role in the results we obtained.

However, the main point of our manuscript is not the new materials. The main point and the novelty is that we have realized a photodetector and the performance we obtained with it. We disagree with the statement that similar measurements have been reported in Refs. 18 and 25. In these excellent works, photoemission was investigated, not detection via absorption of microwave photons. With the risk of stretching the analogy, measuring the properties of a semiconductor laser or light emitting diode are not similar to analyzing a photodiode.

Moreover, in Refs. 18 and 25, the efficiency in converting electrons tunneling through the DQD to emitted photons is more than two orders of magnitude smaller than the efficiency we report for the reverse process, the conversion of incident photons to electrons tunneling through the DQD. This demonstrates, contrary to what one naively would guess, that the underlying mechanisms for photon absorption and emission are different. We emphasize this point in the manuscript.

The main measurements of the current experiment, reported in Figure 3, is not well explained. In a photon assistant transport process (PAT), one expects to observe current in the DQD charge stability diagram in a region a finite detuning, parallel to the DQD interdot region (zero detuning) where the resonant condition is matched. But the current distribution observed in Fig.3(a) does not follow this typical expected PAT pattern. It is instead a broad continuum of current (also out of resonance), not just along the dashed region depicted by the authors for positive and negative detuning. Do the authors understand this behavior or are able to model it? If so, adding this to the manuscript could help the reader understanding the process.

Reply: Yes, we do understand this. The response along the constant detuning lines is discontinued by the source and drain tunneling thresholds. The detuning dependence is also now discussed and is in line with theory: a finite photoresponse is obtained out of the resonance condition because of linewidth-broadening due to decoherence.

The features in the photoresponse in the experiment matches to the theory on a quantitative level as illustrated in the updated manuscript. For more details, we also refer to questions #8 and #9 by the first Referee.

Furthermore, I have the following concerns:

1) It is not clear in the manuscript if the authors are able to estimate the charge decoherence rate for electrons confined in the DQD. On page 3 the authors write: 'The effect of the DQD is captured by a change in linewidth given by an additional loss term $\kappa_{\text{DQD}} = 4g^2/\Gamma$ as an additional channel opens up for the photons to escape the resonator', but it is not immediately clear what ' Γ ' represents. As indicated in the original theoretical proposal in Ref. {4}, this term should also contain a contribution coming from the intrinsic decoherence of the DQD (see for example Eq.18 of the cited work). For a sufficient fast dot-lead tunnel rate one could neglect the intrinsic DQD decoherence. The DQD used in this work presents quite high charging energy and it is used a finite detuning (far away from the sweet spot). Therefore, it should present a substantial decoherence rate, usually higher than a GHz. Instead, the typical tunneling rates reported in this manuscript are of few tens of MHz. This is an important point that would be worth clarifying in the manuscript.

Reply: We thank the Referee for these critical and at the same time constructive remarks. First, we agree with the Referee that the denominator in κ_{DQD} includes an intrinsic DQD decoherence term. We have now updated the manuscript including both the internal DQD dephasing and relaxation rates, see our response point #16 to Referee 1 above.

Second, the observation by the referee that a decoherence rate larger than the tunnel rates is to be expected in our system motivated us to revisit our data analysis and to include both DQD dephasing and relaxation explicitly in the theoretical model. This extensive effort resulted in a modified parameter estimation scheme as well as an inclusion of additional measurement results in order to be able to determine all parameter values. We now include a fit to the phase response data of Fig. 3b to determine the coupling g , the DQD relaxation rate Γ_r , as well as the total decoherence rate $\tilde{\Gamma}$. With this fitting scheme we now obtain $\tilde{\Gamma}=790$ MHz, $g/2\pi = 21$ MHz and $\Gamma_r = 23$ MHz. These parameter values are in line with what has been previously reported for similar structures, as pointed out by the Referee.

We have updated the manuscript at appropriate places to clarify the modified fitting scheme and to discuss the obtained parameter values and the resulting consequences for the physical interpretation. Although our main results, in particular the quantum efficiency of 6 %, as well as the quality of our earlier fits, are largely unaffected by the revision, we strongly feel that the description of the system and the comparison to the existing literature has been considerably improved by these modifications.

2) At the end of the manuscript the authors write: ‘Moreover, the demonstrated efficiency is two orders of magnitude higher than the one for photoemission in similar hybrid cavity structures [18, 25]. The reason is that the key mechanism suppressing the photoemission efficiency, DQD relaxation via spurious emission of phonons, is too slow to affect the photodetection process which relies on fast electron tunneling out of the excited DQD.’ This sentence should be supported by measurements. Did the author estimate the relaxation and dephasing rates for their DQD tuned in the studied configuration? Usually, at finite DQD detuning the relaxation rate gets suppressed and the main source of decoherence is represented by $1/f$ charge noise-induced dephasing.

Reply: As a result of the additional analysis discussed in the previous response point, the statement that the “the key mechanism suppressing the photoemission efficiency, DQD relaxation via spurious emission of phonons, is too slow to affect the photodetection” has been revised. The text now reads “*The low emission efficiency is explained by the dominance of DQD relaxation over photoemission, occurring at rates Γ_r and κ_{DQD} respectively. For photodetection, the DQD relaxation instead competes with tunneling out of the DQD (rate Γ) which occur at comparable rates in our device.*” As also discussed in the previous response point, this sentence is now supported by measurements; the DQD relaxation and dephasing rates are determined in the updated fitting scheme.

3) It will be very informative and important for the manuscript to provide an estimate of the relaxation and dephasing rates for the DQD charge system and to compare those rates with the electron tunneling rates across the DQD (Γ in the manuscript).

Reply: We fully agree with the Referee and have, as discussed in the two previous response points, obtained both the DQD relaxation and dephasing rates within our updated parameter estimation scheme. We have also compared the obtained rates with the electron tunneling rates. The manuscript has been revised to provide this information and corresponding discussion.

4) I do not find trivial the sentence ‘The second and the third condition have a trade-off because $g \propto t$ ’. Could the authors briefly describe (in the manuscript) why $g \propto t$?

Reply: The effective interaction strength g between the DQD and resonator is proportional to the interdot tunnel coupling t , as is discussed in e.g. Ref. [18]. Formally, this dependence arises in the Jaynes-Cummings Hamiltonian (Eq. 3 in the Theory section, in Methods) when transforming from the local, left-right basis to the global, binding-anti binding basis. The proportionality $g \propto t$ can be understood physically in the following way: The microwave field in the resonator couples to the electric

dipole moment of a single electron, tunneling between the two dots. The stronger the inter-dot tunnel coupling t is, the faster the electron goes back and forth in the DQD and hence, the stronger is the dipole moment.

To provide additional information on the dependence $g \propto t$, we reformulated the sentence to “*The second and the fourth condition have a trade-off because a small inter-dot tunnel coupling suppresses interaction with the resonator as $g \propto t$, and therefore reduces κ_{DQD} .*” We also added a reference, as requested by the first Referee (response point 19), which points to a more detailed, in-depth description of this dependence.

5) The resonance condition can be realized for different interdot tunneling rate (t) and for different DQD-detunings. How this impacts the quantum efficiency of the photon detection process?

Reply: The referee is correct; the resonance condition can be achieved for different t and δ . To achieve high efficiencies, a large directivity p_f is desirable. That corresponds to a situation where the excited (ground) state is pre-dominantly localized on the right (left) dot, promoting electrons entering from the left and leaving to the right, as required for photo-detection. As the directivity is given by

$$p_f = \frac{\delta_r}{hf_r} = \sqrt{1 - (2t/hf_r)^2},$$

a large directivity is reached for low inter-dot tunneling rate t . However, as discussed in the previous response point, reducing the tunneling rate reduces the effective photon-electron coupling g , resulting in the above-mentioned trade-off.

To further clarify this point we modified the discussion after our Eq. (2) in the manuscript.

6) The estimated coupling strength $g=3.6$ MHz is quite low. I guess this is the coupling strength at some finite detuning (renormalized by the mixing angle). It would be useful to quote also the bare coupling strength (estimated for zero DQD-detuning) in order to directly compare it with other studied similar hybrid devices. Do the authors think that the detection efficiency would benefit by promoting the system in the strong coupling regime?

Reply: We agree with the referee that a value $g/2\pi = 3.6$ MHz is quite low. As discussed in the response point above, we made a renewed data analysis based on an updated theoretical model. This resulted in larger estimated value of $g/2\pi = 23$ MHz, which is better in line with the values reported in previous works.

We also agree that it would be useful to present the bare coupling strength in the manuscript, to allow a more clear-cut comparison to other works. The relation between the effective coupling g and the bare coupling g_0 is $g = g_0 \cdot 2t/h f_r$, such that $g = g_0$ at zero DQD-detuning as the Referee points out. This gives a bare coupling of $g_0 = 38$ MHz. We added this information to the manuscript and provide the relation between g and g_0 in the methods section.

To answer the last question of the referee, we point out that to reach unit efficiency, we require $\kappa_{DQD} = \kappa$. A higher coupling strength thus allows us to reach unit efficiency with higher bandwidth and/or higher decoherence rates. To provide this information, we have extended the text in the manuscript below Eq. (2), where the conditions for close-to-unit efficiency are discussed.

7) It is not completely easy to follow the reasoning of the authors in explain how they estimated the microwave power at the detector. For example, it seems that they assume 'identical' lines (the 3 lines reported in Fig.1 of suppl. Material). With this assumption and by measuring the signal power reflected for a signal frequency well detuned from the resonator, they try to estimate the extra attenuation introduced by the cable. This assumption is not true in general. The transmission input and RF output are not identical (there is the amplifier in the output line, so at least 2 more SMA connectors). In general, even if the 2 lines would have built nominally identical, there are always impedance mismatches introduced by connectors and by the different rf components (for example usually a cryogenic amplifier presents a VSWR not =1). Furthermore, this attenuation can also be frequency-dependent. Therefore, estimating the attenuation far detuned from the resonator fundamental mode can introduce some miscalibration. In addition, from the transmission line, there is another set of bonding wires (to the left port of the resonator) that the microwave photons do not probe while sent through the RF reflection (R) port.

Reply: We thank the referee for pointing this out. We have added details on the calibration frequency, measurements and included now an in-depth description of the microwave power estimation to the supplemental material. We present data showing that the response is within 20 % accuracy constant in the calibration frequency interval (containing also the photodetector frequencies). This removes the concern of the frequency dependence that the referee raised. We also provide now estimates of the impact of the extra connectors, bonding wires and reflections at the microwave components and explicitly state that this implies a 20 % relative accuracy for the quantum efficiency number, that is: $\eta = 6 \% \pm 1 \%$.

Reviewers' Comments:

Reviewer #1:

Remarks to the Author:

The authors have adequately responded to most of my previous comments and I find their arguments persuasive. The manuscript has been improved after the proofreading. However, I think that the readability of the manuscript should be enhanced before publication. The DQD relaxation rate is reported but it is not clear how its value is obtained from the fit. In Supplemental Material, the authors show the stability diagram and the photocurrent plots taken at different days. In order to help the reader to compare different sets of data, it would be useful to display, in the same figure, a direct comparison between charge stability diagram, photocurrent and phase plots for data taken at the same day or with the same set of plunger gate voltages. Concerning the reversal of the current polarity for $\Delta=0$ (point 8), the comment given by the authors appears rather speculative. First of all, I note that this effect is not visible in Fig. 3d and f in Supplemental Material. Additionally, in ref Cornia et al., Scientific Reports 9, 19523 (2019) rectification effects have been observed at high microwave power (>-55 dBm) while in the present case the microwave power is in the fW range.

Reviewer #2:

Remarks to the Author:

I appreciate the authors for detailed explanations on the three comments in my previous report. However, unfortunately, I could find no answer/revision to the most essential point: what the authors call "efficiency" in this manuscript is "photon-to-electron" conversion efficiency and is not "detection" efficiency. For an efficient photodetector, the electric current induced by a single microwave photon can be discriminated from background noise, clearly and with high probability. In other words, evaluation/estimation of "electron-to-signal" conversion efficiency is inevitable. In case of superconducting-qubit-based detectors, the high qubit readout fidelity was the key for efficient microwave detection.

According to the authors, their device does not have single-photon sensitivity but can be called as "photodetector", because there are existing devices called photodetector in spite of the lack of single-photon sensitivity. I disagree with this point due to the following reasons. (i) In the community of microwave photodetection, researchers are developing microwave "single-photon" detector for quantum information processing, so "microwave photodetector" implicitly means a device with single-photon sensitivity (refs 27-30). (ii) By reading the first paragraph, where the authors emphasize the significance of "single photon" detector, most readers are naturally misled that a microwave single-photon detector has been developed here. The authors should state clearly that their device lacks the single-photon sensitivity at the early stage of this manuscript.

The authors state in the last sentence of the first paragraph that "Our results pave the way for photodiodes with single-shot microwave photon detection, at the theoretically predicted unit efficiency". However, as far as I could check reference 4, what reaches unit efficiency is not the detection efficiency but the photon-to-electron conversion efficiency. Therefore, I feel that this statement would be misleading.

If one does not require the single-photon sensitivity, commercial spectrum analyzers and vector network analyzers are nothing but the continuous microwave detector, following the authors' definition. Then, the authors' claim that this is the "first" realization of "efficient and continuous" detection in the microwave regime seems inadequate. Comparison of figures of merits between the author's device and the commercial ones would be informative to readers.

To summarize, distinction between the photon-to-electron conversion efficiency and the actual detection efficiency is not clearly stated in this manuscript which may mislead most readers. I therefore do not recommend publication of this manuscript in this journal.

REVIEWER COMMENTS

Reviewer #1 (Remarks to the Author):

The authors have adequately responded to most of my previous comments and I find their arguments persuasive. The manuscript has been improved after the proofreading. However, I think that the readability of the manuscript should be enhanced before publication.

Reply: We thank for the positive assessment of our work. Below we address each of specific points raised by the referee.

The DQD relaxation rate is reported but it is not clear how its value is obtained from the fit.

Reply: We thank the Referee for pointing out that this needs to be clarified. As mentioned now in the updated manuscript, the parameters $\tilde{\Gamma}$, and Γ_r are collectively obtained from fitting the data shown in Fig. 3 b, c, and d. In the updated manuscript, we correct this by stating: "*by fitting the data shown in Fig.3b, c and d, we extract [...] the DQD relaxation rate of $\Gamma_r = 23$ MHz that, in addition to the other parameters, determines the low power photocurrent and the quantum efficiency of Fig. 2.*"

In Supplemental Material, the authors show the stability diagram and the photocurrent plots taken at different days. In order to help the reader to compare different sets of data, it would be useful to display, in the same figure, a direct comparison between charge stability diagram, photocurrent and phase plots for data taken at the same day or with the same set of plunger gate voltages.

Reply: Yes, it is surely good to have the measurements timewise as close as possible to each other. We tried to do this as best as we could within required measurement time. Figures 3e and 3f in the supplemental material together with Fig. 3b of the main article in fact form such set with minor drifts between the measurements. We extended the caption of Fig. 3 of the supplemental material with "*Panels e and f present a similar set of measurements carried out at day 1 which is the same day when the phase response measurement of Fig. 3b of the main manuscript was done. These three plots form a set of data where the charge stability diagram (panel e here), photocurrent (panel f here) and phase plot (panel 3b of the main article) are measured with the same gate voltages within a few mV drifts at most between the measurements.*" to point this out.

Concerning the reversal of the current polarity for $\delta=0$ (point 8), the comment given by the authors appears rather speculative. First of all, I note that this effect is not visible in Fig. 3d and f in Supplemental Material. Additionally, in ref Cornia et al., Scientific Reports 9, 19523 (2019) rectification effects have been observed at high microwave power (>55 dBm) while in the present case the microwave power is in the fW range.

Reply: We agree fully that this statement is speculative. That was in fact our purpose in order not to give the impression that we know the origin for sure but rather to point to a possible origin. Since the origin of this feature is not important for the main claims of the manuscript, a precise understanding is not critical. But it definitely is an interesting feature to be studied in the future. To be clear about that our statement indeed is a bit speculative, we modify the corresponding text in the supplementary to “*We cannot for sure identify the origin of the current reversing response at $\delta=0$ of Fig. 3a in the main article. However, we speculate that it arises from a similar effect as reported in Ref. 7.*”

Regarding the effect not being visible in Fig. 3d and f of the supplemental material, we already commented that in the earlier version of the supplemental material with the phrases “*Panels d and f have a stronger response to the positive current polarity only around $\delta=0$ as compared to the data in Fig. 3a of the main article. This arises as in these measurements the offset voltage of the current pre-amplifier was not tuned as well as for the data in the main article.*” That is: we understand the difference and have even produced it intentionally with a better tuning of the offset of the preamplifier.

Reviewer #2 (Remarks to the Author):

I appreciate the authors for detailed explanations on the three comments in my previous report. However, unfortunately, I could find no answer/revision to the most essential point: what the authors call "efficiency" in this manuscript is "photon-to-electron" conversion efficiency and is not "detection" efficiency. For an efficient photodetector, the electric current induced by a single microwave photon can be discriminated from background noise, clearly and with high probability. In other words, evaluation/estimation of "electron-to-signal" conversion efficiency is inevitable. In case of superconducting-qubit-based detectors, the high qubit readout fidelity was the key for efficient microwave detection.

Reply: From the comments of the reviewer, we feel we are largely stuck in a discussion on semantics. We agree with the reviewer that what we “*call "efficiency" in this manuscript is "photon-to-electron" conversion efficiency*””. We have edited thoroughly the beginning of the manuscript to make this clear. However, we do not agree with the reviewers claim that this efficiency cannot be called a “detection efficiency” because efficient detection would imply that “*a single microwave photon can be discriminated from background noise, clearly and with high probability*”. In other words, we don’t think it makes sense to argue that “photon detection” is by definition the same as “single photon detection”. In fact, as we discussed in our previous response, the way we use the term “detection efficiency” is common in the field of photodetectors in the optical regime.

Fortunately, this largely semantic issue is not important for the results presented in our work. To remove any source of confusion for the reader, we have therefore followed the recommendation of the editor and edited the beginning of the manuscript in order to clarify that we discuss the

efficiency of photon-to-electron conversion. Any discussion about the possibilities to reach efficient single photon detection in our system is deferred to the Discussion section at the end of the manuscript.

According to the authors, their device does not have single-photon sensitivity but can be called as "photodetector", because there are existing devices called photodetector in spite of the lack of single-photon sensitivity. I disagree with this point due to the following reasons. (i) In the community of microwave photodetection, researchers are developing microwave "single-photon" detector for quantum information processing, so "microwave photodetector" implicitly means a device with single-photon sensitivity (refs 27-30). (ii) By reading the first paragraph, where the authors emphasize the significance of "single photon" detector, most readers are naturally misled that a microwave single-photon detector has been developed here. The authors should state clearly that their device lacks the single-photon sensitivity at the early stage of this manuscript.

Reply: As pointed out in our response to the previous point, we disagree with the reviewer that our device cannot be called a photodetector. Let us explicitly comment on the two points raised by the reviewer. (i) The reviewer claims that, differently from photodetectors in the optical regime, a "*microwave photodetector*" *implicitly means a device with single-photon sensitivity*". The field of photodetection in the microwave regime is still in its infancy and to make claims about suitable terminology, in fact different from the optical regime, is not meaningful. (ii) As discussed in the previous response point, to avoid that the readers are misled we have reformulated the beginning of the manuscript.

The authors state in the last sentence of the first paragraph that "Our results pave the way for photodiodes with single-shot microwave photon detection, at the theoretically predicted unit efficiency". However, as far as I could check reference 4, what reaches unit efficiency is not the detection efficiency but the photon-to-electron conversion efficiency. Therefore, I feel that this statement would be misleading.

Reply: We agree with the reviewer that what we call "efficiency" in this manuscript is "photon-to-electron conversion efficiency". However, this does not make the cited statement misleading. First, the "photon-to-electron conversion efficiency" corresponds to the well known "quantum efficiency" that is used to describe established devices such as CCD photodetectors and solar cells. Second, with the modifications of the manuscript discussed above we feel we have removed any risk for misunderstandings regarding the meaning of detection efficiency. Third, to reach a near-unity single-shot microwave photon detection efficiency we would need to extend our device with real time single-charge detection, as described in the manuscript and also pointed out by the reviewer above. To emphasize that such single charge detection is already experimentally demonstrated we add a citation to S. Gustavsson, M. Studer, R. Leturcq, T. Ihn, K. Ensslin, D. C. Driscoll, and A. C. Gossard, Phys. Rev. Lett. 99, 206804 (2007) (Ref. #4 in the updated version).

If one does not require the single-photon sensitivity, commercial spectrum analyzers and vector network analyzers are nothing but the continuous microwave detector, following the authors' definition. Then, the authors' claim that this is the "first" realization of "efficient and continuous"

detection in the microwave regime seems inadequate. Comparison of figures of merits between the author's device and the commercial ones would be informative to readers.

Reply: The referee claims that following our definition, commercial spectrum and network analyzers could, just as our device, be considered as microwave photodetectors. We strongly disagree with this statement, for several reasons:

- First and foremost, commercial spectrum analyzers and vector network analyzers work typically by (a possible frequency down conversion of the wave followed by) measuring the amplitude and/or phase of the (lower frequency) signal. These instruments therefore probe the wave properties of the signals and not, like our device, their particle, or photon, properties.

- Second, and related to the first point, since there is no process in these instruments where incident photons with some probability are converted to conduction electrons, any discussion of a detection efficiency, similar to our device, becomes meaningless.

- Third, since these instruments do not probe the photon properties of the incoming microwaves, there is no possibility to employ them for efficient single microwave photon detection. This is again different from our device, as discussed in the previous response point.

To summarize, distinction between the photon-to-electron conversion efficiency and the actual detection efficiency is not clearly stated in this manuscript which may mislead most readers. I therefore do not recommend publication of this manuscript in this journal.

Reply: As is clear from our response above, we feel that with the performed updates of the manuscript, the risk for misleading the readers has been eliminated.

To comply with our article templates, the text must be split into:

- Introduction (ideally 1000 words or less), which must include the background and rationale for the work. The final paragraph should be a brief summary of the major results and conclusions. The results of the current study should only be discussed in this final paragraph.

- Results, which must be split into subheaded sections, ensuring that the subheadings are no longer than 60 characters including spaces.

- Discussion, without subheadings.

- Methods, which must be split into subheaded sections, ensuring that the subheadings are no longer than 60 characters including spaces. There is no word limit for this section.

The whole article (excluding methods and abstract) should ideally not be longer than 5000 words.

Figure Guidelines:

1. All figures should have a title briefly describing the whole figure. Figure titles should ideally be no longer than about one line, with minimal symbols and no punctuation. It would be ideal if you could limit the use of acronyms in figure titles. Any acronym used should be defined in the caption.

2. Panels should be individually labelled and referred to in the caption. Do not refer to panels via their position, as this may change in the production of the final pdf.

3. We have a 350 words limit on figure captions, therefore feel free to expand them to suitably and comprehensively describe what the readers are looking at. Ideally, figures should be as self-consistent as possible, without the need to refer to the text to grasp their meaning.

4. Please supply figures so that every element of each figure is editable (i.e. we can highlight and edit the text, and move individual parts of the figures around). When making these changes please ensure resolution stays high at 300dpi.

5. The meaning of all error bars (sd? Sem?) and how they were calculated should be described within the captions of all figures in which they occur.

6. Please ensure that all plotted variables are accompanied by units of measures, unless dimensionless. Please use (a.u.) if arbitrary.

Reply: We added the subheadings and reorganized the abstract and introduction content to comply these requirements.

Supplementary Information Guidelines:

Supplementary Information should be provided as a separate pdf file. The first page should be a cover page containing only the title in the form "Supplementary Information - Title of the Manuscript" and the authors in the form "Smith et al.", with no affiliations.

The text in the Supplementary Information should be organised using the following subheaded sections:

- Supplementary Notes, headed "Supplementary Note 1 - *title*", "Supplementary Note 2 - *title*", etc.
- Supplementary Discussion
- Supplementary Methods
- Supplementary References, which should be self-contained. That is, references mentioned in both the main text and the Supplementary Information should be part of both reference lists so that the Supplementary Information does not refer to the reference list in the main paper and vice versa. References cited in the Supplementary Information should be numbered sequentially from 1. They should be formatted in the same style as the main paper.

When referring to Supplementary Information from other parts of the main manuscript, it would be ideal if you could refer each time to the specific Supplementary Item (e.g. "See Supplementary Note 1" or "as highlighted in Supplementary Figure 3") rather than generically to "Supplementary Information".

Also, keep in mind that:

- Supplementary Figures, should be labelled and referred to as Supplementary Figure 1, not Fig 1, throughout both the Supplementary Information and the main text
- Supplementary Tables should be labelled and referred to as Supplementary Table 1, not Table 1, throughout both the Supplementary Information and the main text
- Any equation throughout the Supplementary Information should be numbered from 1, not from S1. They should be cited in the text as "Supplementary Equation XX".

Reply: We changed the formatting of the supplemental material to comply these guidelines.